# Using Sum-Product Networks to Assess Uncertainty in Deep Active Learning

**Mohamadsadegh Khosravani**                                          *mko041@uregina.ca*
*Department of Computer Science, University of Regina*

**Sandra Zilles**                                                     *sandra.zilles@uregina.ca*
*Department of Computer Science, University of Regina*

**Reviewed on OpenReview:** *https://openreview.net/forum?id=Ai9XpjGxjl*

## Abstract

The success of deep active learning hinges on the choice of an effective acquisition function, which ranks not yet labeled data points according to their expected informativeness. Many acquisition functions are (partly) based on the uncertainty that the current model has about the class label of a point, yet there is no generally agreed upon strategy for computing such uncertainty. This paper proposes a new and very simple approach to computing uncertainty in deep active learning with a Convolutional Neural Network (CNN). The main idea is to use the feature representation extracted by the CNN as data for training a Sum-Product Network (SPN). Since SPNs are typically used for estimating the distribution of a dataset, they are well suited to the task of estimating class probabilities that can be used directly by standard acquisition functions such as max entropy and variational ratio. The effectiveness of our method is demonstrated in an experimental study on several standard benchmark datasets for image classification, where we compare it to various state-of-the-art methods for assessing uncertainty in deep active learning.

## 1 Introduction

The large amount of data required for training deep neural networks motivates the use of active learning (Settles, 2009), in which the learning method actively selects unlabeled data points whose labels are then acquired. Selecting particularly informative data points reduces the number of data points required for producing an accurate predictor. Uncertainty-based approaches consider a point informative if its class label is highly uncertain given the current training data (Lewis & Gale, 1994). To reduce the risk of sampling many redundant points, active learning often uses mixed selection criteria, combining uncertainty with distribution-based or diversity-based strategies. In deep active learning, be it with purely uncertainty-based or mixed approaches, one challenge is to estimate uncertainty. MC Dropout (Gal & Ghahramani, 2016) estimates the uncertainty of the label of a point $x$, for a CNN trained on a given dataset. It uses random dropout to create many (similar but not identical) variants of the CNN and then averages the probabilities output by these variants on point $x$, in order to get a robust estimate of the posterior probability of any given class label for $x$. This probability estimate can then be fed into a standard uncertainty function like entropy, variational ratio, or mutual information, in order to assess uncertainty. MC Dropout showed very good performance on some image datasets, but it was recently reported to be of little effect on more complex datasets like CIFAR-10 (Pinsler et al., 2019).

The main contribution of this paper is a new method for estimating posterior probabilities of class labels, using an approach that is conceptually very simple, yet more effective than MC Dropout on complex data. We deploy the probabilistic nature of Sum-Product Networks (SPNs (Poon & Domingos, 2011))—deep graphical models that handle probabilities by design, typically with the purpose to estimate the data distribution. Their structure carries more intuitive semantics than that of neural networks. Due to their probabilistic nature,

SPNs are particularly suited to the task of deriving probabilities needed for uncertainty-based deep active learning. In a nutshell, our method trains a CNN and then uses its extracted data representation in order to train an SPN which is used to assess the probabilities with which the training data allows us to assign unseen data points to classes. As with MC Dropout, these probability estimates are then fed into standard functions to calculate uncertainty.

Due to its simplicity, our method is widely applicable, both in a standalone uncertainty-based active learning method (which we call SPN-CNN), and in the uncertainty component of mixed deep active learning methods. For example, BatchBALD (Kirsch et al., 2019) is a method that combines uncertainty (assessed via MC Dropout) with a diversity-based criterion. One can replace the MC Dropout component in BatchBALD with our method to assess uncertainty, resulting in a method we call SPN-BatchBALD. In a similar vein, our method can be plugged into various deep active learning tools.

One limitation is that our method only works well when the number of classes in the underlying classification problem is not too large. In our experiments, it performed well for standard datasets with 10 classes, but not for datasets with 100 or more classes. This, however, is not a limitation of specifically our method, but a limitation of uncertainty assessment as such (and thus of all methods trying to assess uncertainty), as we explain in the beginning of Section 5.

We empirically evaluate the accuracy of our method both in standalone uncertainty-based sampling, and in a mixed setting, on standard benchmark datasets for image processing. In the standalone setting, we compare SPN-CNN to MC Dropout, using two standard uncertainty-based acquisition functions (max entropy and variational ratio). The mixed setting compares SPN-BatchBALD to BatchBALD, using mutual information for acquisition. It turns out that SPN-CNN outperforms MC Dropout in all cases except when deployed with Max Entropy on the least complex tested dataset. In our experiments, it even outperforms the state-of-the-art method Bayesian Batch (Pinsler et al., 2019) (which aims to counteract redundancy), except in one case when a very large initial training data set was provided. (Arguably though, active learning is more critical when initial training datasets are not very large.) When compared to BatchBALD, again SPN-BatchBALD is significantly more accurate.

Further, we mimic the experimental setup from a study that proposes a standalone method for assessing uncertainty in CNNs (Lakshminarayanan et al., 2017) (devoid of a concrete application such as active learning). Our results strongly suggest that our assessment of uncertainty alleviates the well-known problem of overly confident predictions made by deep neural networks. The SPN component detects that the trained CNN is uncertain on test data points that differ substantially from the training data, while the CNN itself expresses overly high confidence in its predictions. MC Dropout, by contrast, does not detect the predictive uncertainty of the CNN to a comparable extent. In addition, an analysis of the expected calibration error is included, which shows that the confidence estimates of SPN-CNN are well calibrated with the accuracy of the predictions, i.e., SPN-CNN has a low risk of being too confident in its predictions.

In sum, training an SPN on the features extracted by a CNN is simple, easy to implement, and yet highly effective in estimating probabilities for assessing uncertainty in active learning.

## 2 Related Work

The two main approaches by which an active learner selects data points are uncertainty sampling (choosing a point for which the class label is most uncertain in the current model) (Lewis & Gale, 1994; Nguyen et al., 2022) and diversity sampling (choosing points to reflect the underlying distribution of the data) (Brinker, 2003). In uncertainty sampling, standard measures of uncertainty are, e.g., Shannon Entropy (Shannon, 1948) and Variational Ratio (Freeman, 1965); both of these measures need estimates of the probability that a data point $x$ has class label $c$, given the training data. Such "class probability" estimates can be derived in various ways, e.g., from the output of a neural network after applying softmax, or from the proximity of a point to the decision boundary when using a linear separator like an SVM (Brinker, 2003).

Deep active learning has become a very active branch of research (Ren et al., 2021; Zhan et al., 2022), driven mostly by the need to reduce the typically large amounts of data required to train deep classification models. Deep model structures to which active learning has been applied include stacked Restricted Boltzmann

Machines (Wang & Shang, 2014), variational adversarial networks (Sinha et al., 2019), as well as CNNs (Wang et al., 2016; Gal et al., 2017). Uncertainty sampling is a popular approach here as well (Gal et al., 2017; Yi et al., 2022; Ren et al., 2021; Zhan et al., 2022), with MC Dropout (Gal et al., 2017) as a well-known state-of-the-art method. One criticism of uncertainty sampling is that it tends to result in highly correlated queries (Sener & Savarese, 2017), which may prevent methods like MC Dropout from scaling well to large datasets. In fact, it was noted that MC Dropout works very well on the MNIST dataset, but has a poor performance on the more complex CIFAR-10 data (Pinsler et al., 2019). This problem can be addressed with alternate selection principles or with the combination of uncertainty-based and alternate principles.

Several deep active learning methods use coresets in sample selection, instead of uncertainty. When optimizing an objective function on sets of points, a coreset of a set $P$ of points is a subset of $P$ whose optimal solution approximates that for $P$. For actively training CNNs, this kind of sparse approximation is done by computing geometric center points for clusters of labeled points and then selecting unlabeled points whose distance to the nearest center is maximal (Sener & Savarese, 2017). Bayesian Batch (Pinsler et al., 2019) also uses coresets for actively training CNNs, but in a Bayesian rather than a geometric setting.

Other methods use uncertainty in different ways, to create more cost-efficient methods. One example of such an approach for training CNNs assigns pseudo-labels for large batches of data points with high classification confidence (Wang et al., 2016). These pseudo-labels are used for re-training the model, which can result in a more effective process of selecting from the remaining unlabeled data points. Some methods combine self- or semi-supervised methods with standard active learning methods. For instance, the method by Yi et al. (2022) performs a pretext task (like rotation prediction) over the pool of unlabeled data points, which is then divided into batches based on the performance on the pretext task. Then, uncertainty-based active sampling is performed over one batch. By comparison, the method by Siméoni et al. (2021) performs first an unsupervised feature learning task and then semi-supervised tasks in cycles.

Assessing the uncertainty of deep neural network predictions is of interest also outside the context of active learning (Abdar et al., 2021). Overly confident predictions of CNNs are a common problem that negatively impacts their real-world applicability (Lakshminarayanan et al., 2017). Therefore, not all methods for assessing uncertainty are suitable for active learning. For example, the method proposed by Lakshminarayanan et al. (2017) uses ensembles of deep neural networks, combined with adversarial training, in order to evaluate the uncertainty of model predictions. This method was shown to be effective when the ensemble size is sufficiently large, which likely makes it infeasible in terms of runtime in a deep active learning setting.

In light of this related work, the novelty of our method lies in its way of estimating class probabilities: they are entirely derived from the probabilistic estimates produced by an SPN (Poon & Domingos, 2011) trained on the data representation extracted by the CNN model. SPNs are often used for density estimation and have, to the best of our knowledge, not been applied to derive class probabilities for uncertainty sampling. An alternative method for assessing uncertainty could be a Gaussian Process, but it would have to be deployed with a deep kernel, which results in a cubic runtime complexity for inference (compared to linear inference time for SPN) as well as a cubic training time. Hence, we consider SPNs better suited to the efficient and effective assessment of uncertainty in deep active learning.

## 3 Preliminaries

We assume an initial training dataset $\mathcal{D}_{train} = \{(x_i, y_i)\}_{i=0}^N$ of pairs $(x_i, y_i)$, where $x_i$ is a data point and $y_i \in C = \{c_1, \ldots, c_{|C|}\}$ is the corresponding class label from a fixed set $C$ of labels. Active learning means that, in addition, the learning algorithm has access to a pool $\mathcal{D}_{pool}$ of unlabeled data points. After processing the initial training dataset, the learning algorithm selects a small set of points from the pool, which will then be labeled by an oracle. The newly labeled set is added to the training set, and the model is retrained.

The goal of active learning is to train a high quality predictive model with less data than would be required for passively learning a model of similar predictive performance. The criterion for selecting points from the pool is usually given via an acquisition function $a(x, \mathcal{M})$, which takes as input the current model $\mathcal{M}$ and a point $x$ from the pool, and assigns an informativeness value to $x$. The acquisition strategy is then to select

a maximally informative pool point:

$$x^* = argmax_{x \in \mathcal{D}_{pool}} a(x, \mathcal{M}) \tag{1}$$

Acquisition functions can be based on various aspects, e.g., the distribution of the pool or the uncertainty with which the model makes predictions on individual pool data points. Here we focus on the latter type of acquisition function; in our experimental study, we work with three of the most popular uncertainty-based acquisition functions, namely Max Entropy, Variational Ratio, and BALD.

**Max Entropy** expresses uncertainty as predictive entropy (Shannon, 1948).

$$\mathbf{H}[x, \mathcal{D}_{train}] = -\sum_y p(y \mid x, \mathcal{D}_{train}) \log p(y \mid x, \mathcal{D}_{train})$$

**Variational Ratio** models certainty as the confidence in predicting the most likely class $y$ for $x$ (Freeman, 1965).

$$\mathbf{varra}[x, \mathcal{D}_{train}] = 1 - \max_y p(y \mid x, \mathcal{D}_{train})$$

**BALD** (Houlsby et al., 2011) uses mutual information to measure uncertainty.

$$\mathbf{I}[x, \mathcal{D}_{train}] = \mathbf{H}[x, \mathcal{D}_{train}] - \mathbf{E}_{p(\omega \mid \mathcal{D}_{train})} \mathbf{H}[x, \omega, \mathcal{D}_{train}]$$

The effectiveness of acquisition functions largely depends on how accurately the probability $p(y \mid x, \mathcal{D}_{train})$ can be approximated. Some common types of models, like logistic regression or deep neural networks, provide a probability value $p(y \mid x, \omega)$ (where $\omega$ refers to the model) as their output when making a prediction for $x$; this could be directly used as a surrogate for $p(y \mid x, \mathcal{D}_{train})$ in an acquisition function. While this method is simple and straightforward to implement, it may be highly susceptible to individual model parameters and thus not yield a good approximation of $p(y \mid x, \mathcal{D}_{train})$. In the neural network case, this problem has been addressed by the MC Dropout approach (Gal & Ghahramani, 2016) explained below. Our main contribution is an alternate approach to approximating $p(y \mid x, \mathcal{D}_{train})$ for deep active learning, based on SPNs.

**MC Dropout**  Dropout is a regularization method in deep learning, usually applied during training (Srivastava et al., 2014). MC Dropout (Gal & Ghahramani, 2016) repeatedly uses dropout at test time, to create $T$ versions $\omega_t$, $t = 1, \ldots, T$ of the network model $\omega$, and then estimates $p(y \mid x, \mathcal{D}_{train})$ by averaging the values of $p(y \mid x, \omega_t)$. This average is less reliant on individual parameters of $\omega$ than the probabilities $p(y \mid x, \omega)$ directly output by the model $\omega$ when it makes predictions.

**SPNs**  An SPN is a directed acyclic graph with layers of nodes of different types (leaf, sum, and product nodes) (Poon & Domingos, 2011). Leaf nodes take as input values of random variables $X^1, \ldots, X^d$; in our context, each random variable refers to one component of a $d$-dimensional data point $x_i$. The value of a sum node is a weighted sum of the values of its children, where the weights assigned to the children sum up to 1. The value of a product node is the product of its children's values. The value of the root can then be interpreted as the SPN's prediction of the probability with which data point $x_i$ belongs to a specific class $c \in C$. (For classification problems with $|C|$ classes, one has to deploy $|C|$ such DAGs.) Similar to a neural network, an SPN is determined by (a) its graphical structure, and (b) its parameters, which are the weights of the children of sum nodes. However, their interpretation is purely probabilistic.

SPNs can be learned in various ways. Random SPN (Peharz et al., 2018) constructs a random region graph, which is then populated with tensors of SPN nodes. Other methods construct SPNs based on convolutional product layers (Wolfshaar & Pronobis, 2020; Butz et al., 2019). (In our experiments, we use the structure chosen by Wolfshaar & Pronobis (2020).) In order to learn the weights, standard optimization procedures for minimizing a loss function given the training data are used, see, e.g., (Peharz et al., 2018).

In an SPN, each node in a product layer represents a distribution over a disjoint set of variables, while each node in a sum layer corresponds to a mixture of distributions. The root represents the joint distribution of the leaves. Thus, an SPN is suitable for modeling real-world phenomena of a probabilistic nature.

SPNs lend themselves to performing inference tasks, such as marginal or conditional queries, for multiple reasons. First, SPNs perform inference in linear time (Zhao et al., 2015), while competing probabilistic

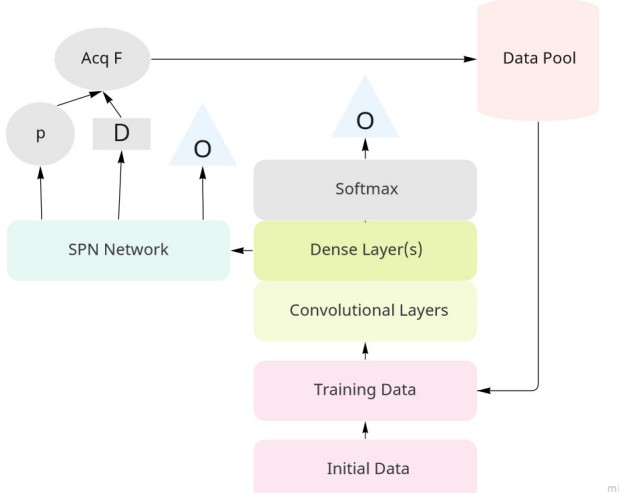

Figure 1: SPN-CNN deep active learning approach.

models, such as Bayesian networks and Markov networks, have an inference time exponential in the model size. Second, unlike typical deep neural networks, SPNs can handle variable input and output sizes for marginal and conditional queries (Kalra et al., 2018).

## 4 SPN-Based Uncertainty Sampling

SPNs are deep probabilistic models, which are mostly used for the purpose of density estimation. Therefore, they are an apt model for extracting the distribution of a dataset. The main idea of our deep active learning approach is to use SPNs for extracting probability estimates that can be used in standard uncertainty-based acquisition functions when actively learning a deep neural network classifier. To this end, we introduce a novel technique for applying SPNs.

Suppose one has trained a different type of model (e.g., a CNN) on the observed data, and extracted a new feature representation of the data. If the observed data is complex, or contains redundancies or noise, then the representation extracted from the data is expected to better capture the distribution of the data. Thus, if an SPN is trained in order to capture the true data distribution, it might be more useful to train it on previously extracted features than to train it on the original data. Formally, instead of approximating $p(y \mid x, \mathcal{D}_{train})$, we use an SPN to approximate $p(y \mid z, \mathcal{D}_{train})$, where $z$ is the representation of $x$. In practice, we take the output of the final dense layer (before the output layer) of a trained CNN and consider it as a feature representation of the input data, on which the SPN is trained. The probability estimates computed by the SPN can then be used as $p(y \mid x, \mathcal{D}_{train})$ values in a specific acquisition function $a$, such as max entropy, variational ratio, or BALD, which then results in the acquisiton strategy as shown in (1).

This SPN-based technique is the basis for our proposed model architecture for deep active learning, called SPN-CNN. As depicted in Figure 1, the three main components of our architecture are (i) a CNN, (ii) an SPN, and (iii) an uncertainty-based acquisition function (like variational ratio or max entropy). The training process works as follows.

1. An initial CNN model is trained on a given set of initial training data points.

2. An initial SPN is trained on the output of the final dense layer of the obtained CNN. Specifically, for each initial training data point $x_0$, the output of the dense layer is fed to the SPN, together with the class label of $x_0$. The data $x_0$ itself is not fed to the SPN.

3. The SPN makes predictions on pool points, which allows us to calculate estimates of $p(y \mid x, \mathcal{D}_{train})$ values as needed in uncertainty-based acquisition functions.

4. A new datum $x'$ is selected via the acquisition strategy in (1), and the class label $y'$ of $x'$ is obtained from the oracle. (This step may be repeated a predefined number of times to collect a batch of new labeled data points.)

5. The CNN block is retrained using the (batch of) newly acquired training data $(x', y')$; the corresponding outputs of the dense layer are used to retrain the SPN.

6. If the sample budget is used up then stop, else go to 3.

The output of the SPN (obtained from applying softmax to the vector of prediction values of the individual DAGs; depicted as "O" in Figure 1) is needed when training the SPN, but it is not used by the acquisition function itself. (However, it is used in selecting data points, in order to calculate the weights by which the acquisition function are multiplied.)

Our model has two classifiers: a CNN trained over observed data, and an SPN trained over the CNN's representation of data. Both of them can be used for predicting class labels, as indicated by the two triangles labeled "O" in Figure 1. When evaluating our approach empirically, we consider only the option, in which the final prediction is the output of the SPN. The other option turned out worse in our empirical study and is not reported here.

Training the SPN over the feature representation rather than the original data has advantages that we have tested empirically. In the literature, the highest reported accuracy of an SPN model, trained on the full predefined MNIST training set of size $50K$, is around 98% (Peharz et al., 2018; Wolfshaar & Pronobis, 2020). By comparison, SPN-CNN obtained an accuracy of 97.6% (std deviation 0.045%), trained on only 1000 data points—a promising result.

## 5 Experimental Analysis

We tested our approach both for pure uncertainty-based sampling (SPN-CNN) and for mixed sampling. For the latter case, we use BatchBALD (Kirsch et al., 2019), which is a mixed sampling method that uses the BALD acquisition function but iteratively selects data points to be sampled in a batch, in a way so as to avoid redundancy and ensure sufficient diversity. By SPN-BatchBALD we denote a variant obtained when replacing BatchBALD's component for assessing class probabilities $p(y \mid x, \mathcal{D}_{train})$ (which is based on MC Dropout) by our SPN-based method.

Our evaluation uses four standard benchmark datasets: MNIST (Deng, 2012), CIFAR-10 (Krizhevsky et al., accessed May 2022), Fashion-MNIST (Xiao et al., 2017) and SVHN (Netzer et al., 2011). All our experiments used the same SPN structure, but a different CNN structure for each dataset. Details on the experimental setup, such as the SPN and CNN architectures, hyperparameters, off-the-shelf tools used, dropout procedures, etc., are provided in Appendix A.

**Choice of datasets** We deliberately exclude datasets like CIFAR-100 and ImageNet, which target classification into 100 or more classes. They do not seem suited to a study on assessing uncertainty of class labels, since label uncertainty arguably becomes very hard to distinguish when the number of object classes becomes very high. When assigning a confidence value for an individual data point $x$ to each of $|C|$ classes, the maximum such confidence value will typically be smaller when $|C|$ is larger. This gives uncertainty estimators a smaller range to operate in, and thus will intuitively make uncertainty estimators harder to distinguish in terms of their effectiveness. In a small study on CIFAR-100, none of the uncertainty sampling approaches that we tested performed better than random selection of samples. To the best of our knowledge, all publications focusing on uncertainty in CNNs use datasets with many classes only as binary class datasets. For example, Lakshminarayanan et al. (2017) use ImageNet but convert it into a two-class dataset. Variational Adversarial Active Learning (Sinha et al., 2019) is also tested on datasets with many classes, yet its uncertainty component does not relate to the class probabilities of these classes, but to the uncertainty of an internal binary prediction problem (classifying data points as "labeled" or "unlabeled").

### 5.1 Pure Uncertainty-Based Sampling

We compared our method to four others, which differ in the way data points are actively sampled. Two simple baselines train the same initial CNN as our method but actively sample as follows: (i) *Random* samples uniformly at random without replacement from the pool; (ii) *Deterministic* uses standard acquisition functions, where class probabilities are simply the outputs of the CNN before applying softmax. Two state-of-the-art competitors that we tested are (iii) MC Dropout (Gal et al., 2017) and (iv) Bayesian Batch (Pinsler et al., 2019); this way, we compare SPN-CNN to another uncertainty sampling method as well as to a method using an entirely different approach. Note that SPN-CNN, as well as *Deterministic* and MC Dropout, compute class probabilities to be used in acquisition functions; we tested both Max Entropy and Variational Ratio as acquisition functions.

**How to read our plots** Plots in Figures 2 through 4 have the number of training data points (initial training set size plus actively sampled points) on the $x$-axis and accuracy on the $y$-axis. The $y$-intercept of a curve equals the accuracy of a model trained only on the initial training data; each subsequent point represents the accuracy after processing one additional batch of acquired points. Each curve is the average over five runs (std dev is shown in Appendix B), with randomly chosen training, validation, and pool sets per run and per method. Note that the curves for SPN-CNN and for Bayesian Batch often have substantially different $y$-intercepts than the other pure uncertainty-based methods, since they correspond to outputs from other model structures; the remaining methods produce outputs based on the same CNN structure and thus have similar $y$-intercepts.

**MNIST** All competing methods were given the same test data, which is the predefined test set consisting of $10K$ data points. For each run and each method, we sample other data sets from the predefined $50K$ training data points as follows: the $50K$ points are shuffled at random, $40K$ points are randomly chosen for the pool, $5K$ points for the validation set, and from the remaining $5K$ points we randomly choose an initial training set of only 20 points. Acquisition was performed 98 times, each time picking 10 points (for a total of $1K$ training points in the end), following the respective acquisition strategy and then updating the model. Figure 6 shows the resulting accuracy, averaged over five runs. For both acquisition functions, SPN-CNN performs better than the two baselines and than Bayesian Batch. After five batch acquisitions with Max Entropy, MC Dropout surpasses SPN-CNN, but with Variational Ratio it is as bad as the *Random* baseline. Note that, in the first few acquisitions, *Random* performs as well as the other methods (even better than *Deterministic*), but it cannot keep up later on. One possible explanation is that in the first steps active learning gains little to no advantage over passive learning, but when the model becomes more refined, active sample selection makes a bigger difference.

This experiment also demonstrates the potential strength of an SPN when trained over the feature representation of data. In the literature, the highest reported accuracy of an SPN model, trained on the full predefined MNIST training set of size $50K$, is around $98\%$ (Peharz et al., 2018; Wolfshaar & Pronobis, 2020). By comparison, SPN-CNN obtained an accuracy of $97.6\%$ (std deviation $0.045\%$ ), trained on only 1000 data points. Hence, training SPNs on the representation of data seems promising.

**CIFAR-10** CIFAR-10 requires a more complex CNN structure (we used VGG (Simonyan & Zisserman, 2014)) and larger training set sizes than MNIST. The predefined test set of $10K$ points was used in all evaluations. We performed two tests. The first one is similar to the setups used by Siméoni et al. (2021) and Yi et al. (2022), with $1K$ initial training data points and nine acquisitions in batches of size $1K$. The second one (performed only for Max Entropy) used $5K$ initial training data points and four acquisition batches of 5K points each. In both cases and for each method tested, we shuffle the predefined training set of $40K$ points, then randomly choose a pool of size $30K$ and a validation set of size $5K$; the initial training set is sampled at random from the remaining $5K$ points. The result for the first setting is in Figure 2(a–b). Here SPN-CNN is the winner, followed by *Determinisitic*. Neither MC Dropout nor Bayesian Batch can compete; they behave similarly to *Random*. Figure 2(c) shows the result for the second setting, with larger initial training data sets and larger step sizes. Here, MC Dropout and *Deterministic* are better than *Random*, but the winners are SPN-CNN and Bayesian Batch. Interestingly, Bayesian Batch needs the larger initial training data set/batch sizes to be competitive with SPN-CNN. This suggests that SPN-CNN is particularly effective when data is less abundant.

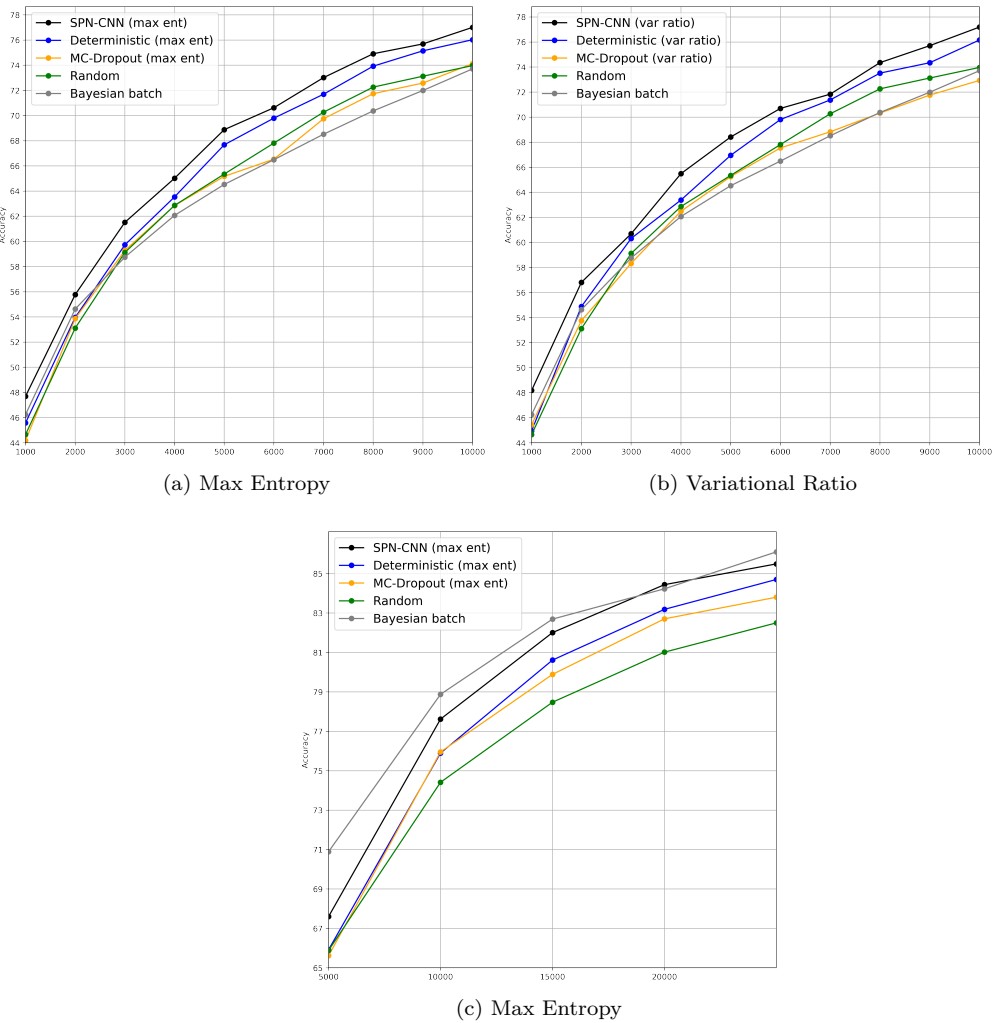

(a) Max Entropy

(b) Variational Ratio

(c) Max Entropy

Figure 2: Accuracy on CIFAR-10, with an initial training set size and acquisition batch size of 1000, for (a) and (b), and with initial training set and batch size of 5000 for (c). Note that plot (c) has a different scale on the $y$-axis than the other two plots.

**Fashion-MNIST**  Here, we use VGG as the CNN structure. The predefined test set has $10K$ points; from the $50K$ given training data points, we randomly sample $40K$ for the pool, $5K$ for validation, and then $1K$ from the remaining ones as initial training data. We performed 9 acquisitions of $1K$ points each. Figure 3(a–b) shows the results averaged over five runs. SPN-CNN outperforms all other methods for both Max Entropy and Variational Ratio. Bayesian Batch is still doing fairly well after the first batch acquisition, but overall both Bayesian Batch and MC Dropout here seem no better than the *Deterministic* baseline.

**SVHN**  This dataset has $73,257$ and $26,032$ data points for training and testing, respectively. $63,257$ data points were chosen randomly as pool set, $5,000$ for validation and $1,000$ as initial training set. All settings for the experiment (model structure, hyperparameters, . . . )  were the same as for CIFAR-10. SPN-CNN outperforms the baselines and MC Dropout in all acquisition steps, see Figure 3(c–d). Again, in the initial steps Bayesian Batch has better performance, but is surpassed by SPN-CNN in the fifth acquisition.

**Run-time comparison**  Table 1 shows run-times in seconds on a local ubuntu Server 20.04.1 with 6-core, 64GB RAM, 128 GB SSD disk drive, GPU NVIDIA Corporation GP100GL [Tesla P100 PCIe 16GB]. For MNIST, we acquired 98 batches of 10 points each plus the training time for initial set with 20 points; for CIFAR-10, Fashion-MNIST, and SVHN, we report the time needed for the acquisition of 9 batches of 1000

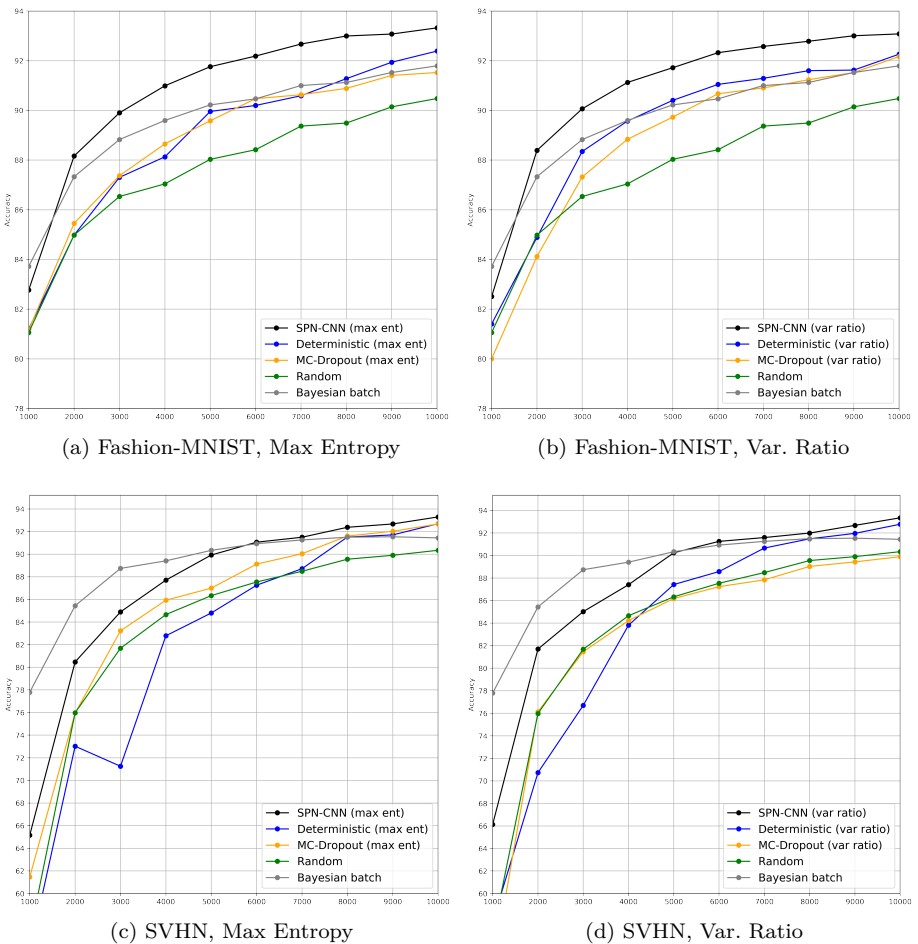

Figure 3: Accuracy on Fashion-MNIST (a–b) and SVHN (c–d).

Table 1: Run-time (in seconds). "Max" is for Max Entropy, "Var" for Variational Ratio.

| Dataset | Random | Deterministic | | Bayesian Batch | MC Dropout | | SPN-CNN | |
| --- | --- | --- | --- | --- | --- | --- | --- | --- |
| | | Max | Var | | Max | Var | Max | Var |
| MNIST | 1,527 | 1,633 | 1,617 | 10,603 | 2,111 | 2,277 | 5,321 | 5,288 |
| CIFAR-10 | 4,072 | 4,058 | 4,062 | 16,709 | 5,800 | 6,231 | 6,539 | 6,404 |
| Fashion-MNIST | 3,920 | 3,952 | 3,984 | 32,447 | 6,254 | 6,997 | 10,241 | 10,242 |
| SVHN | 3,980 | 3,992 | 4,020 | 28,445 | 5,552 | 5,506 | 8,871 | 8,358 |

points each plus the training time for an initial set with 1000 points. In all run-time experiments, we trained the SPN in 650 epochs. Not surprisingly, random acquisition requires the least time. Overall, SPN-CNN's run-time is up to 2.5 times that of MC Dropout, yet only around 30%–50% of that for Bayesian Batch.

**Statistical tests** We applied paired z-tests, where each pair in the comparison is a pair of mean accuracy values for a test data set of 10,000 points, averaged over 5 runs. We collected $p$-values for each of ten acquisition steps with batch size 1000. On the CIFAR-10 data, both when using variational ratio and when using max entropy, SPN-CNN beat MC Dropout with $p = 0.05$ after 5 batch acquisitions, with $p = 0.01$ after 6 and after 8 batch acquisitions, and $p < 0.01$ after all other batches. Further, SPN-CNN with variational ratio beats Bayesian Batch on CIFAR-10 with $p = 0.05$ after 1 batch, $p = 0.01$ after batches 2 and 6, and $p < 0.01$ for all other batches. With max entropy, SPN-CNN beats Bayesian Batch on CIFAR-10 with $p = 0.04$ after 1 batch, $p = 0.01$ after batches 2 and 6, and $p < 0.01$ for all other batches. For FashionMNIST,

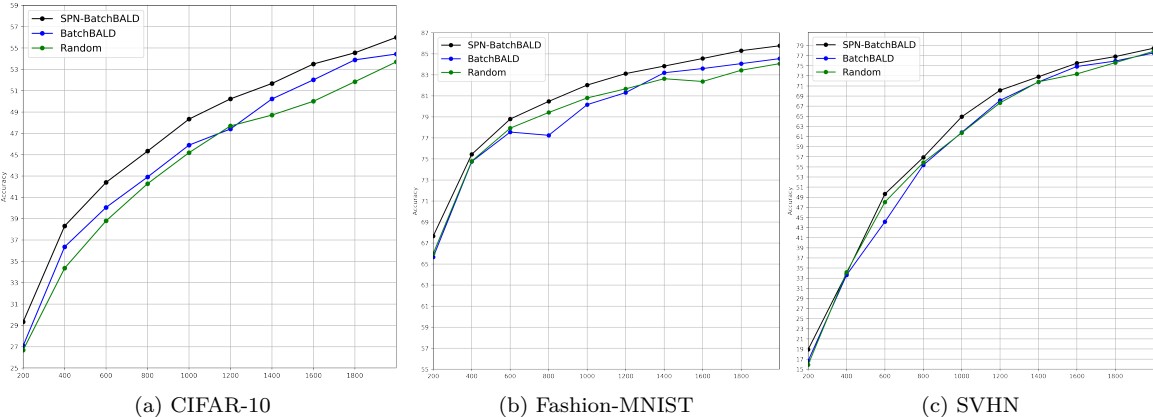

Figure 4: Accuracy on various datasets. Note that the scales on the *y*-axes differ.

the results are similar. SPN-CNN max entropy beats MC Dropout always with $p < 0.01$; with variational ratio, batches 1 and 10 had $p$-values of 0.02 and 0.01, respectively, all other batches had $p < 0.01$. Compared to Bayesian Batch, SPN-CNN cannot win for the first batch, but wins highly significantly from batch 2 onwards: with variational ratio, the $p$-value at batch 2 is 0.1, and it is smaller than that from batch 3 onwards; with max entropy, we get $p = 0.04$ at batch 2, $p = 0.01$ at batch 3, and $p < 0.01$ with every batch afterwards. These results confirm our claim that SPN-CNN significantly beats both MC Dropout and Bayesian Batch on CIFAR-10 and Fashion-MNIST. (Note that our $p$-values, listed in Appendix D, are small enough to withstand Holm-Bonferroni corrections for multiple tests on the same data.)

## 5.2 Mixed Sampling Methods

In terms of running time, BatchBALD does not scale very well to large batch sizes. Therefore, we used an initial training set size and acquisition budget size of 200 in all our experiments on SPN-BatchBALD and BatchBALD. We compared these two methods on CIFAR-10, SVHN and Fashion-MNIST, with the same *Random* baseline as in the previous experiments. All other setup details are identical to those reported for the pure uncertainty sampling case. Again, we report averages from five runs, see Figure 4. In all cases, SPN-BatchBALD is the clear winner. We confirmed the significance of our results with statistical tests; details on standard deviation and statistical tests are reported in Appendix D.

## 5.3 Assessing Overconfidence Via Out-of-Distribution Examples

Lakshminarayanan et al. (2017) proposed a line of research focusing on only uncertainty assessment, outside the context of any specific application like active learning. Their method is not practical for deep active learning, as it requires training a whole ensemble of CNNs; empirically, it seems that at least five CNNs in the ensemble are required for satisfactory performance. We therefore do not evaluate this method, but use the experimental setting from (Lakshminarayanan et al., 2017) in order to compare SPN-CNN to MC-Dropout and *Deterministic*. In this setup, uncertainty is assessed on out-of-distribution examples from unseen classes. Thus, an uncertainty estimator is considered good if its predicted uncertainty values increase substantially when the test data differs significantly from the training data (Lakshminarayanan et al., 2017).
All three tested approaches estimate class probabilities, which can then be plugged into an uncertainty function. Like Lakshminarayanan et al. (2017), we use entropy as the uncertainty function. For each approach separately, we train a model on SVHN and then test it on two test data sets: (i) the usual SVHN test data, (ii) the usual CIFAR-10 test data. Finally, Figure 5 shows density curves for the entropy values resulting from the class probability predictions on these two test sets. For SPN-CNN, a substantial difference in entropy between the two test sets is observed: on SVHN data, predictions have consistently very low uncertainty, while the uncertainty is overall much larger and more spread when it comes to the unfamiliar CIFAR-10 test set. MC Dropout and *Deterministic*, by contrast, seem overly confident even when making

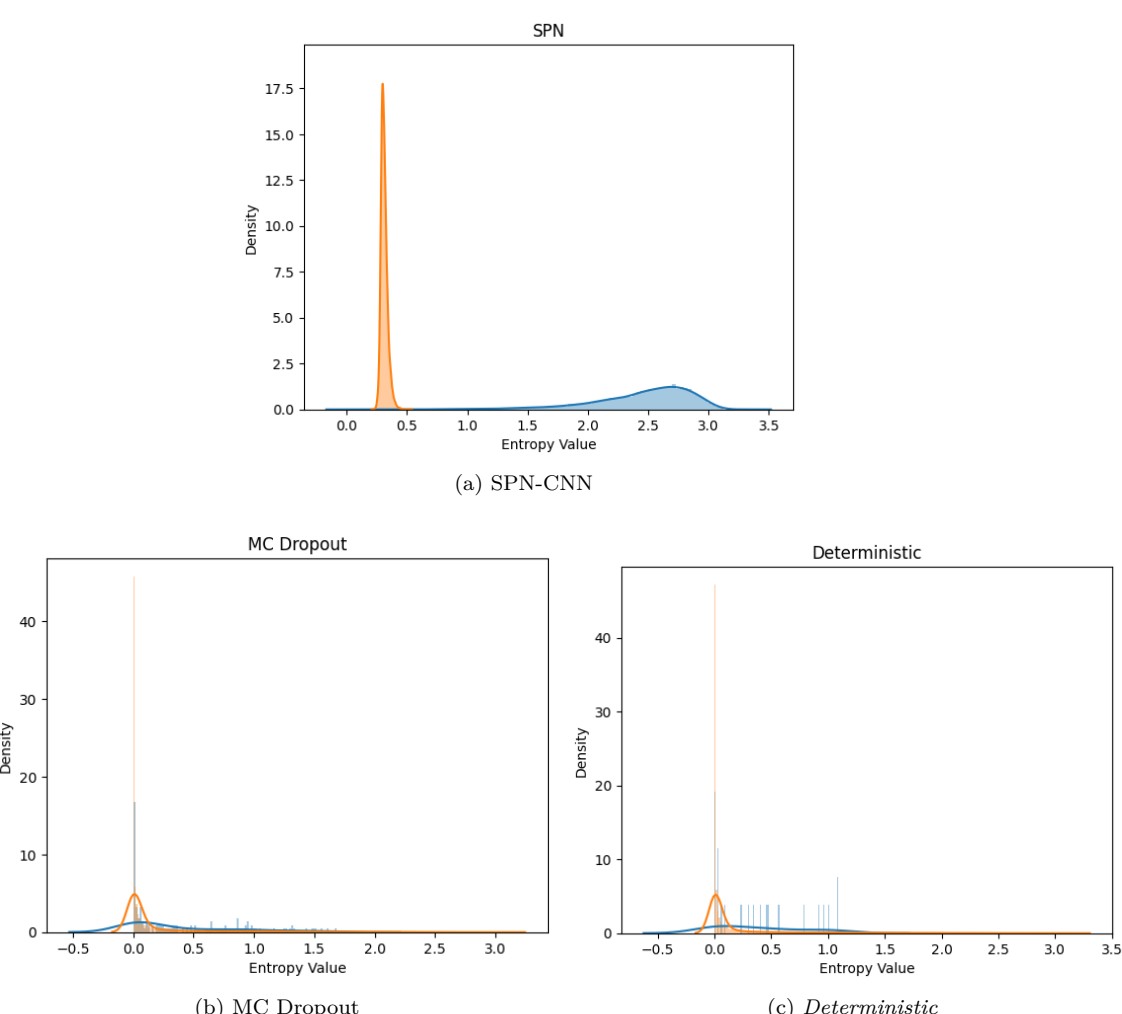

Figure 5: Density of entropy values on SVHN test data (orange) and CIFAR-10 test data (blue) after training on SVHN. Note that the scales on the *y*-axes differ.

predictions on CIFAR-10 data. Lakshminarayanan et al. (2017) noted that overly confident predictions are a major drawback in most deep learning methods, and are problematic in real world applications. Our experiment suggests that the SPN-CNN approach can alleviate this problem for CNNs much better than MC Dropout can; experiments with more data sets would be helpful to further confirm this observation.

### 5.4 Assessing Overconfidence Via Expected Calibration Error

In deep learning, another way to assess overconfidence of predictors is the Expected Calibration Error (ECE), which measures the expected difference between the accuracy and confidence of the trained model. The lower ECE, the better calibrated a model is. Various methods have been proposed to reduce ECE, with temperature scaling being among the most prominent ones (Guo et al., 2017). However, such methods to reduce ECE can only be applied after training the model. This means, they cannot directly be used to calibrate uncertainty estimates during the course of uncertainty-based active learning, since the active training of the model requires uncertainty estimates at every acquisition step.

Our proposed way of assessing uncertainty can be used as a standalone method for uncertainty estimation, but also within deep active learning. Therefore, it is important for our method to be well calibrated even without calibration methods like temperature scaling. We hence computed ECE of our uncertainty estimates in

Table 2: ECE (%) on CIFAR-10 with three different networks.

|  | Uncalibrated CNN | Uncalibrated SPN-CNN | Calibrated CNN | Calibrated SPN-CNN |
|---|---|---|---|---|
| **VGG** | 8.21 | 5.13 | 1.73 | 2.00 |
| **LeNet** | 4.33 | 4.50 | 1.55 | 3.01 |
| **ResNet** | 7.60 | 5.54 | 1.78 | 2.86 |

comparison to the uncertainty estimates of a standard neural network model (without the SPN component). In addition, we compared ECE for both models after applying temperature scaling.

We conducted this experiment with three different model architectures (VGG, LeNet, and ResNet) on the CIFAR-10 data. The models were trained on a random subset of 45,000 data points out of the standard training set of 50,000 points. For calibration with temperature scaling, the remaining 5,000 data points from the standard training set were used as validation set. Finally, ECE was calculated with 15 bins on the 10,000 test data points. This setup is identical to the one used by Kull et al. (2019).

The results are reported in Table 2, demonstrating that SPN-CNN does not produce highly uncalibrated uncertainty estimates, compared to the corresponding CNN alone. After applying temperature scaling, our method (SPN-CNN) is not quite as well calibrated as the method without SPN. However, of importance for applications in active learning, ECE values for the uncalibrated SPN-CNN tend to be lower than for the uncalibrated CNN. This suggests that, in settings where calibration through temperature scaling is not possible since uncertainty estimates are required during model training, the risk of overconfidence is lower for SPN-CNN than for CNN.

## 6 Conclusions

We proposed an appealingly simple method for uncertainty sampling in active learning with CNNs, which estimates class probabilities with an SPN trained over the representation extracted by the CNN. It can be used both as a standalone sampling approach (SPN-CNN) and in combination with other sampling techniques (e.g., SPN-BatchBALD).

In our experimental study, in terms of accuracy, SPN-BatchBALD outperforms BatchBALD, and SPN-CNN is not outperformed by any of the tested competitors, with two exceptions: (i) MC Dropout outperforms SPN-CNN on MNIST when using Max Entropy, and (ii) Bayesian Batch, while mostly worse than SPN-CNN, performs slightly better or comparable for the large training data set and batch sizes we tested on CIFAR-10; large initial training sets however are a less likely situation in the context of active learning. Empirically, there are some weaknesses of MC Dropout from which SPN-CNN does not suffer: Firstly, MC Dropout is sensitive to the dropout rate. If the dropout rate is too small, the model is close to deterministic overall and we will not get useful posterior probability estimates. If the dropout rate is too high, the model will tend to suffer from underfitting. Secondly, our empirical analysis suggests that MC Dropout is more sensitive to the batch size than SPN-CNN (see experiments with batch size 1000 vs. batch size 5000.) Thirdly, empirically in our study, it appears that MC Dropout does not perform well on complex datasets (it performs well on MNIST, but poorly on FashionMNIST and close to randomly on CIFAR-10). This confirms previous reports that MC Dropout is not very effective on more complex data. Apparently, Bayesian Batch requires large initial training sets to compete with SPN-CNN. In sum, our method to estimate class probabilities with an SPN trained on a CNN's feature representation appears to be very effective, at a reasonable run-time cost compared to state-of-the-art methods, and with a simple and easy to implement design.

More experiments should be performed to test our method in other applications of deep learning, e.g., NLP, and with other network architectures, e.g., autoencoders. As discussed above, like other methods for assessing uncertainty in multi-class prediction, our method is not useful when the number of classes becomes too large. In deep active learning with uncertainty sampling, all uncertainty estimators that we tested (including ours) were worse than random acquisition on the CIFAR-100 dataset, which has 100 classes. For the 10-class version, uncertainty-based methods turned out to be suited very well.

## Acknowledgements

This work was supported by the Natural Sciences and Engineering Council of Canada, through the Discovery Grants program and the Canada Research Chairs program. Further, we acknowledge support from CIFAR, through a Canada CIFAR AI Chair held by S. Zilles at the Alberta Machine Intelligence Institute (Amii).

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

# A    Details on Experimental Setup

This section provides details on the setup of the experiments described in Section 5 of the main paper.

## A.1    SPN

For implementing the SPN, we use the lib-spn keras library which is inherited from keras. The SPN structure used in all our experiments is Deep Convolutional SPN (Wolfshaar & Pronobis, 2020). In addition, we employed a dropout layer. We always trained our SPN model over 650 epochs, using the Adam optimizer, and with sparse categorical entropy as loss function.

All parameters except dropout rate were tuned on CIFAR-10 only, and simply adopted for the other datasets in our experiments. The dropout rate was tuned for each dataset separately. Our general tuning procedure was to first fix the learning rate to .005 and the dropout rate to .1, and then perform the following two steps:

- **Number of components**: We tried $16, 32$ and $64$, and finally selected $16$.

- **Number of sum nodes**: We tried the following sequences $[16, 32, 32, 128, 128]$, $[16, 32, 32, 64, 64]$, $[16, 32, 32, 64, 64]$, $[64, 64, 128, 128, 256]$, $[32, 32, 64, 64, 128]$, and picked $[16, 32, 32, 64, 64]$.

This resulted in the following structure (from leaf layer to root layer): leaf layer (with 16 components for each variable) to which we apply dropout, followed by a product layer, sum layer (with 16 sum nodes), product layer, sum layer (with 32 sum nodes), product layer, sum layer (with 32 sum nodes), product layer, sum layer (with 64 sum nodes), product layer, sum layer (with 64 sum nodes), root layers (one for each class). Since the values returned by the root layers were very small, we additionally applied softmax at the last step. We also tried to add a normalization layer before the last sum layer, but did not adopt it since it did not affect the performance substantially.

After fixing structural parameters, the learning rate was manually tuned to .08. To this end, we tried out learning rate values from .005 to .1 in steps of size .01, with a fixed dropout rate of .1. We chose the learning rate that maximized average accuracy on the validation set over two runs.

Finally, we tuned the dropout rate for each dataset separately, trying values from .05 to .5 in steps of size .05. Again, we chose the value that maximized average accuracy on the validation set over two runs. Moreover, we tried dropout in layers other than the leaf layer but this always decreased the accuracy; our reported results were obtained with dropout applied only in the leaf layer. The final value for dropout rate was .3 for Fashion-MNIST and CIFAR-10, and .05 for MNIST.

## A.2    CNN for MNIST

For all methods except Bayesian Batch, we use a structure similar to LeNet (LeCun et al., 1998): two convolutional layers (filter size 32), maxpool layer, two convolutional layers (filter size 64), dropout layer (dropout rate .5), flatten layer, dense layer (128 neurons), dropout layer (dropout rate .5), dense layer with

softmax activation. The output of the dense layer is reshaped ($16 \times 8$) to use as input for the SPN. We used the SGD optimizer with a learning rate of .001 and as loss function we chose categorical cross entropy. The number of epochs was set to 100, with a batch size of 120.

We set hyperparameters by assessing accuracy on the validation set for several values of the hyperparameters. For the learning rate we tried .001 to .1 with step size of .001, each with a dropout rate of .25 and a dropout rate of .5. After fixing the learning rate and the dropout rate, we tested three different numbers of neurons for the dense layer (128, 256, and 512) but since this did not seem to affect the accuracy, we chose the smallest of the tested sizes.

For Bayesian Batch, we directly used the authors' code (Pinsler et al., 2019), since their architecture is totally different from the ones used by the other tested methods.

### A.3 CNN for CIFAR-10

For all methods except Bayesian Batch, the CNN structure used for our CIFAR-10 experiments is VGG (Simonyan & Zisserman, 2014), and consists of a convolutional layer (filter size 64), batchNormalization layer, convolutional layer (filter size 64), batchNormalization layer, maxpool layer, dropout layer (dropout rate .2), convolutional layer (filter size 128), batchNormalization layer, convolutional layer (filter size 128), batchNormalization layer, maxpool layer, dropout layer (dropout rate .3), convolutional layer (filter size 256), batchNormalization layer, convolutional layer (filter size 256), batchNormalization layer, convolutional layer (filter size 256), batchNormalization layer, maxpool layer, dropout layer (dropout rate .4), convolutional layer (filter size 512), batchNormalization layer, convolutional layer (filter size 512), batchNormalization layer, convolutional layer (filter size 512), batchNormalization layer, maxpool layer, dropout layer (dropout rate .5), flatten layer, dense layer (with 512 neurons), batchNormalization layer, dropout layer (dropout rate .5), softmax layer with 10 neurons.

The output of the dense layer was reshaped ($16 \times 32$) as input for the SPN. We used SGD with learning rate .001 as optimizer and categorical cross entropy as loss function. The number of epochs was set to 250, with a batch size of 100.

We took the VGG model directly from (Brownlee, 2016). Since the structure was established, we did not attempt hyperparameter tuning as much as with the model used for MNIST. We only tried the learning rates .01 and .001, as well as the dropout rates .25, .3 and .5 for the two last dropout layers. For the number of neurons in the dense layer we tried 256 and 512.

For Bayesian Batch, we directly used the authors' code (Pinsler et al., 2019), since their architecture is totally different from the ones used by the other tested methods.

For BatchBALD, we used the authors' code (Kirsch et al., 2019) for computing the BatchBALD acquisition function. The rest of the code was the same as in our other experiments.

### A.4 CNN for Fashion-MNIST

For all methods apart from Bayesian Batch, the structure and hyperparameter settings were the same as for the CIFAR-10 experiments, except that the learning rate was .0005.

We did not attempt to tune the dropout rate at all (keeping it the same as for the CIFAR-10 data), and tried only three different values for the learning rate (.1, .001, and .0005), picking .0005 as it performed best on the validation set.

For Bayesian Batch, we directly used the authors' code (Pinsler et al., 2019), since their architecture is totally different from the ones used by the other tested methods.

For BatchBALD, we used the authors' code (Kirsch et al., 2019) for computing the BatchBALD acquisition function. The rest of the code was the same as in our other experiments.

Table 3: Accuracy and standard deviation for MNIST over five runs, for 10 selected (equidistant) acquisition steps. This table corresponds to Figure 6.

| Method | 100 | 200 | 300 | 400 | 500 |
|---|---|---|---|---|---|
| SPN-CNN (max ent) | 74.29 ± 1.65 | 86.67 ± .98 | 92.23 ± 1.46 | 94.34 ± 0.63 | 95.31 ± 0.39 |
| Deterministic (max ent) | 60.25± 5.09 | 80.77± 3.08 | 88.28± 2.06 | 91.29± 0.64 | 93.49± 1.28 |
| MC Dropout (max ent) | 76.62± 4.77 | 87.37± 1.93 | 92.58± 1.12 | 94.2± 1.12 | 95.94± 0.50 |
| SPN-CNN (var ratio) | 74.29± 3.64 | 86.67± 1.89 | 92.23± 0.53 | 94.34± 0.57 | 95.31± 0.49 |
| Deterministic (var ratio) | 63.64± 6.80 | 81.25± 2.99 | 87.49± 3.00 | 92.7± 0.96 | 93.62± 0.44 |
| MC Dropout (var ratio) | 74.4±2.97 | 85.26±2.07 | 89.31±2.22 | 92.68± 0.71 | 93.43± 0.36 |
| Bayesian Batch | 9.86±0.69 | 10.20±0.73 | 90.79±1.27 | 93.19±0.76 | 94.46±0.65 |
| Random | 80.06±2.47 | 87.5± 2.06 | 91.14±0.39 | 92.4±0.83 | 93.13± 0.41 |
| Method | 600 | 700 | 800 | 900 | 1000 |
| SPN-CNN (max ent) | 95.51 ± 0.27 | 96.524 ± 0.34 | 96.92 ± 0.34 | 97.13 ± 0.26 | 97.77 ± 0.32 |
| Deterministic (max ent) | 93.47± 1.05 | 95.50± 0.29 | 96.07± 0.45 | 96.61± 0.43 | 97.15 ± 0.18 |
| MC Dropout (max ent) | 96.42± 0.44 | 97.30± 0.25 | 97.77± 0.27 | 97.97± 0.13 | 98.09± 0.16 |
| SPN-CNN (var ratio) | 95.51± 0.31 | 96.52± 0.34 | 96.92± 0.24 | 97.13± 0.18 | 97.77± 0.06 |
| Deterministic (var ratio) | 94.63± 0.98 | 95.57±0.53 | 96.48±0.15 | 97.24±0.27 | 97.52±0.2 |
| MC Dropout (var ratio) | 93.59±0.71 | 94.57±0.24 | 95.03±0.27 | 95.64± 0.40 | 95.75±0.29 |
| Bayesian Batch | 95.28±0.61 | 95.57±0.49 | 96.03±0.26 | 96.33±0.16 | 96.73±0.35 |
| Random | 93.62± 0.28 | 94.52±0.15 | 94.93±0.13 | 95.17± 0.18 | 95.62±0.36 |

### A.5 CNN for SVHN

With the exception of Bayesian Batch, the structure and hyperparameter configurations for the experiments were identical to those used for the Fashion-MNIST experiments. The learning rate was set at .0005.

Again, we directly used the authors' code (Pinsler et al., 2019) for Bayesian Batch, since their architecture is totally different from the ones used by the other tested methods. As in the Fashion-MNIST experiment, we used the authors' code (Kirsch et al., 2019) for computing the BatchBALD acquisition function.

### A.6 Settings for "Uncertainty Alone" Experiment

The CNN and SPN structures and settings for the "uncertainty alone" experiment were identical to those used for the active learning experiment on SVHN, as described in Subsection A.5 in this document. The optimizer was SGD with the learning rate of 0.0005. The only difference compared to the active learning experiment was that the model was trained on the whole training data set.

## B Result Plots with Standard Deviation

This section presents the same plots as in Section 5 of the main paper, but with standard deviation indicated in addition. For the sake of readability, we did not show standard deviation in the plots in Section 5 of the main paper. As one can see here, Figures 6, 7, 10 and 9 are not as easy to analyze as the corresponding figures in the main paper. Therefore, for each of these three figures, we here added a table showing the numerical values of accuracy and standard deviation plotted in the figures. For MNIST, Figure 6 plots accuracy at 100 different $x$-values; here we chose 10 equidistant values for the corresponding table (Table 3). Note that, in accordance with the results reported in the main paper, the standard deviation was calculated over five runs for each dataset.

## C Code

Code that we used off the web is the following

- For MC Dropout we used the code by Gal et al. (2017). We modified the CNN in their code to be the same as the CNN that we use in SPN-CNN, *Random* and *Deterministic*.

- For Bayesian Batch, we used the code by Pinsler et al. (2019).

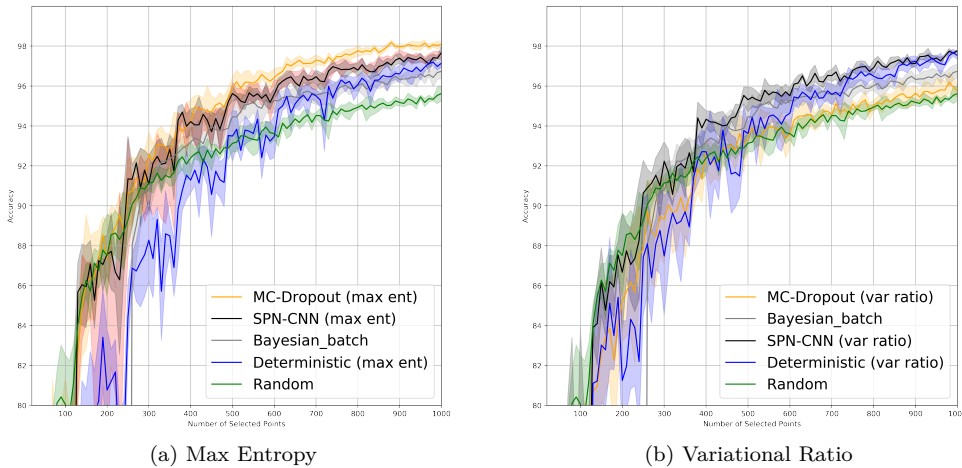

(a) Max Entropy             (b) Variational Ratio

Figure 6: Accuracy results and error bars on the MNIST data, averaged over five runs, with std dev indicated.

Table 4: Accuracy and standard deviation for CIFAR-10 over five runs, after processing 1000, 2000, ..., 10,000 data points (i.e., for the initial training data set and the subsequent 9 batch acquisitions). This table corresponds to Figure 7.

| Method | 1000 | 2000 | 3000 | 4000 | 5000 |
|---|---|---|---|---|---|
| SPN-CNN (max ent) | 48.65 ± 0.66 | 56.09 ± 0.42 | 61.79 ± 0.56 | 65.59 ± 0.35 | 68.25 ± 0.92 |
| Deterministic (max ent) | 45.39 ± 0.83 | 53.65 ± 1.71 | 60.37± 0.74 | 64.52± 0.97 | 66.48± 0.94 |
| MC Dropout (max ent) | 44.19 ± 2.03 | 53.87 ± 0.9 | 59.28± 1.41 | 62.85± 0.65 | 65.17± 1.04 |
| SPN-CNN (var ra) | 47.62 ± 1.36 | 57.18± 1.06 | 61.69± 0.73 | 65.92± 0.27 | 69.27± 0.79 |
| Deterministic (var ratio) | 45.84± 1.01 | 54.98± 1.61 | 60.74± 1.35 | 63.92± 1.13 | 67.76± 1.23 |
| MC Dropout (var ratio) | 45.45±1.31 | 53.75±0.84 | 58.33±0.57 | 62.49± 0.55 | 65.26± 1.6 |
| Bayesian Batch | 46.25±0.65 | 54.63±0.5 | 58.76±0.98 | 62.07±1.11 | 64.53±0.84 |
| Random | 45.92±2.1 | 53.07± 0.6 | 58.28±1.5 | 62.02±1.8 | 66.47± 0.45 |
| Method | 6000 | 7000 | 8000 | 9000 | 10000 |
| SPN-CNN (max ent) | 70.79 ± 0.65 | 72.58 ± 0.54 | 74.96 ± 0.42 | 75.92 ± 0.49 | 77.17 ± 0.69 |
| Deterministic (max ent) | 69.67± 1.02 | 72.09± 1.35 | 72.92± 0.62 | 74.95± 1.1 | 76.16 ± 0.52 |
| MC Dropout (max ent) | 66.54± 1.38 | 69.75± 0.54 | 71.74± 0.62 | 72.59± 0.99 | 74.13± .7 |
| SPN-CNN (var ratio) | 71.18± 1.12 | 73.04± 0.38 | 74.9± 0.7 | 76.28± 0.64 | 77.32± 0.48 |
| Deterministic (var ratio) | 70.06± 0.96 | 71.96±0.72 | 73.4±0.87 | 74.5±0.53 | 76.34±0.55 |
| MC Dropout (var ratio) | 67.55±1.03 | 68.83±0.7 | 70.33±0.84 | 71.76± 0.45 | 72.94±1.02 |
| Bayesian Batch | 66.5±0.62 | 68.5±1.11 | 70.38±1.00 | 71.99±0.96 | 73.71±0.73 |
| Random | 67.45± 1.74 | 69.92±1.13 | 71.08±.99 | 72.6± 1.42 | 74.15±0.72 |

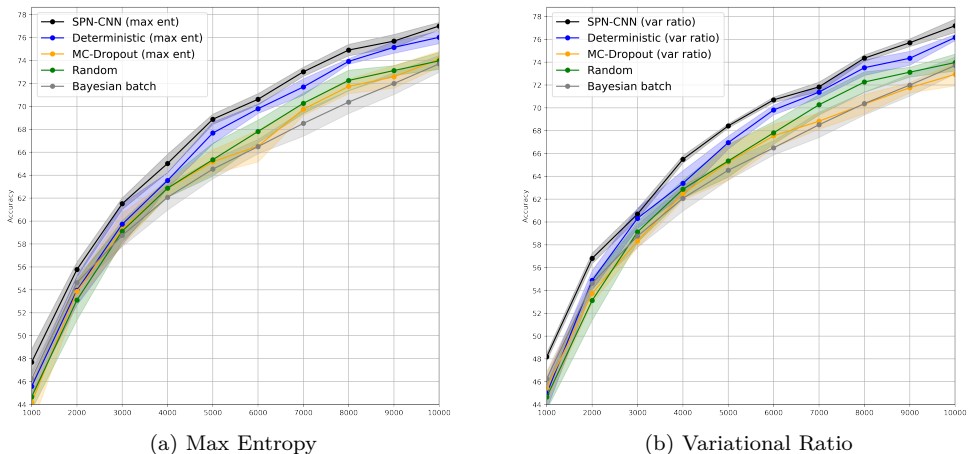

(a) Max Entropy  (b) Variational Ratio

Figure 7: Accuracy results and error bars on the CIFAR-10 data, averaged over five runs, with std dev indicated.

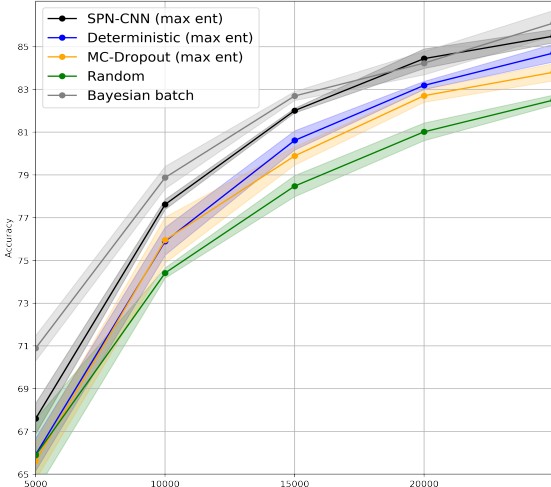

Figure 8: As in Figure 7, but with initial training set size and acquisition batch size of 5000.

Table 5: Accuracy and standard deviation for Fashion-MNIST over five runs, after processing 1000, 2000, ..., 10,000 data points (i.e., for the initial training data set and the subsequent 9 batch acquisitions). This table corresponds to Figure 9.

| Method | 1000 | 2000 | 3000 | 4000 | 5000 |
|---|---|---|---|---|---|
| SPN-CNN (max ent) | 82.38 ± 0.56 | 88.2 ± 0.26 | 90.2 ± 0.11 | 91.07 ± 0.16 | 91.83 ± 0.51 |
| Deterministic (max ent) | 78.68± 0.98 | 85.74± 0.54 | 87.67± 0.56 | 88.85± 0.97 | 89.81± 0.39 |
| MC Dropout (max ent) | 81.29± 0.56 | 85.54± .04 | 87.39± 0.39 | 88.71± 0.63 | 89.66± 0.83 |
| SPN-CNN (SPN, var ratio) | 83.3± 0.2 | 88.06± 0.14 | 90.08± 0.25 | 90.87± 0.29 | 91.51± 0.33 |
| Deterministic (var ratio) | 79.03± 1.92 | 85.24± 0.63 | 87.86± 0.27 | 88.78± 0.39 | 90.22± 0.8 |
| MC Dropout (var ratio) | 80.40±1.23 | 83.85±0.73 | 87.24±0.32 | 88.67± 0.33 | 89.71± 0.46 |
| Bayesian Batch | 83.67±0.28 | 87.35±0.13 | 88.78±0.24 | 89.65±0.30 | 90.23±0.21 |
| Random | 80.25±1.14 | 84.43± 0.40 | 86.23±0.74 | 87.87±0.52 | 88.31± 0.17 |
| Method | 6000 | 7000 | 8000 | 9000 | 10000 |
| SPN-CNN (max ent) | 92.27 ± 0.13 | 92.7 ± 0.16 | 92.9 ± 0.13 | 93.13 ± 0.17 | 93.46 ± 0.12 |
| Deterministic (max ent) | 90.93± 0.11 | 91.57± 0.32 | 92± 0.18 | 91.9± 0.14 | 92.81 ± 0.05 |
| MC Dropout (max ent) | 90.3± 0.51 | 90.57± 0.16 | 90.96± 0.36 | 91.5± 0.12 | 91.54± 0.45 |
| SPN-CNN (var ratio) | 91.66± 0.35 | 92.48± 0.17 | 92.78± 0.27 | 92.84± 0.30 | 93.22± 0.15 |
| Deterministic (var ratio) | 90.47± 0.91 | 90.91±0.40 | 91.67±0.66 | 91.81±0.40 | 92.3±0.37 |
| MC Dropout (var ratio) | 90.58±0.1 | 90.92±0.18 | 91.26±0.11 | 91.53± 0.03 | 92.04±0.18 |
| Bayesian Batch | 90.54±0.1 | 91.03±0.1 | 91.12±0.10 | 91.60±0.17 | 91.84±0.23 |
| Random | 89.16± 0.14 | 89.54±0.80 | 90.05±0.04 | 90.26± 0.99 | 90.71±0.23 |

Table 6: Accuracy and standard deviation for SVHN over five runs, after processing 1000, 2000, ..., 10,000 data points (i.e., for the initial training data set and the subsequent 9 batch acquisitions). This table corresponds to Figure 10.

| Method | 1000 | 2000 | 3000 | 4000 | 5000 |
|---|---|---|---|---|---|
| SPN-CNN (max ent) | 65.15 ± 2.92 | 80.47 ± 2.29 | 84.9 ± 00.93 | 87.70 ± 00.73 | 89.92 ± 00.82 |
| Deterministic (max ent) | 54.86± 6.16 | 73.02± 00.95 | 71.52± 5.50 | 82.78± 00.94 | 84.80± 1.85 |
| MC Dropout (max ent) | 61.46± 2.3 | 75.95± 00.6 | 83.23± 00.78 | 85.92± 00.90 | 86.99± 1.16 |
| SPN-CNN ( var ratio) | 66.13±1.88 | 81.7±1.53 | 85.02±00.84 | 87.41± 00.68 | 90.25± 00.22 |
| Deterministic (var ratio) | 57.48± 2.97 | 70.72± 2.53 | 76.7± 2.57 | 83.81± 1.23 | 87.42± 00.73 |
| MC Dropout (var ratio) | 53.19±4.16 | 76.13±1.89 | 81.45±1.49 | 84.22± 00.56 | 86.18± 00.44 |
| Bayesian Batch | 77.80±00.86 | 85.44±00.25 | 88.74±39 | 89.41±00.19 | 90.34±00.12 |
| Random | 56.34±01.77 | 75.98± 00.87 | 81.65±01.13 | 84.33±00.54 | 86.33± 00.54 |
| Method | 6000 | 7000 | 8000 | 9000 | 10000 |
| SPN-CNN (max ent) | 91.06 ± 00.35 | 91.50 ± 00.38 | 92.37 ± 00.45 | 92.67 ± 00.30 | 93.29 ± 00.30 |
| Deterministic (max ent) | 87.26± 00.6 | 88.72± 1.73 | 91.51± 00.43 | 91.70± 00.59 | 92.69 ± 00.24 |
| MC Dropout (max ent) | 89.12± 00.57 | 90.04± 00.4 | 91.61± 00.08 | 92.03± 00.32 | 92.67±00.21 |
| SPN-CNN (var ratio) | 91.25± 00.25 | 91.6± 00.32 | 92.00± 00.20 | 92.67± 0.38 | 92.35± 00.27 |
| Deterministic (var ratio) | 88.58± 00.87 | 90.66±00.19 | 91.48±00.30 | 91.96±00.33 | 92.77±00.39 |
| MC Dropout (var ratio) | 87.23±00.52 | 87.84±00.52 | 89.03±00.37 | 89.43± 00.29 | 89.84±00.31 |
| Bayesian Batch | 90.92±00.12 | 91.25±00.14 | 91.50±00.17 | 91.53±14 | 91.44±00.00 |
| Random | 87.54± 00.35 | 88.48±00.23 | 89.58±00.30 | 89.9± 00.30 | 90.34±00.30 |

Table 7: Accuracy and standard deviation for CIFAR-10 over five runs, after processing 200, 400, ..., 2000 data points (i.e., for the initial training data set and the subsequent 9 batch acquisitions). This table corresponds to Figure 11.

| Method | 200 | 400 | 600 | 800 | 1000 |
|---|---|---|---|---|---|
| SPN-BatchBALD | 29.33 ± 1.63 | 38.33 ± 1.22 | 42.40 ± 1.18 | 45.37 ± 0.78 | 48.37 ± 1.07 |
| BatchBALD | 27.09 ± 1.83 | 36.36 ± 0.67 | 40.06 ± 0.68 | 42.91 ± 1.43 | 45.90 ± 2.07 |
| Random | 26.69 ± 3.10 | 34.36 ± 1.74 | 38.79 ± 1.77 | 42.28 ± 1.62 | 45.19 ± 1.46 |
| Method | 1200 | 1400 | 1600 | 1800 | 2000 |
| SPN-BatchBALD | 50.23 ± 1.25 | 51.68 ± 0.78 | 53.50 ± 0.62 | 54.55 ± 0.94 | 56.00 ± 0.87 |
| BatchBALD | 47.41 ± 0.71 | 50.22 ± 1.26 | 52.01 ± 0.91 | 53.89 ± 0.19 | 54.44 ± 0.96 |
| Random | 47.70 ± 1.14 | 48.71 ± 1.24 | 50.01 ± 1.13 | 51.84 ± 0.82 | 53.70 ± 0.85 |

Table 8: Accuracy and standard deviation for Fashion-MNIST over five runs, after processing 200, 400, . . . , 2000 data points (i.e., for the initial training data set and the subsequent 9 batch acquisitions). This table corresponds to Figure 12.

| Method | 200 | 400 | 600 | 800 | 1000 |
|---|---|---|---|---|---|
| SPN-BatchBALD | 67.66 ± 1.55 | 75.44 ± 1.73 | 78.78 ± 1.34 | 80.48 ± 1.30 | 82.03 ± 0.47 |
| BatchBALD | 65.66 ± 4.08 | 74.75 ± 1.15 | 77.56 ± 1.81 | 77.25 ± 1.24 | 80.17 ± 1.91 |
| Random | 66.04 ± 2.78 | 74.79 ± 1.40 | 77.93 ± 0.82 | 79.41 ± 0.08 | 80.81 ± 0.55 |
| Method | 1200 | 1400 | 1600 | 1800 | 2000 |
| SPN-BatchBALD | 83.12 ± 0.69 | 83.84 ± 0.39 | 84.56 ± 0.49 | 85.30 ± 0.13 | 85.77 ± 0.34 |
| BatchBALD | 81.31 ± 1.18 | 83.20 ± 0.81 | 83.60 ± 0.33 | 84.07 ± 1.05 | 84.54 ± 1.08 |
| Random | 81.67 ± 0.80 | 82.63 ± 0.80 | 82.34 ± 0.70 | 83.44 ± 0.98 | 84.07 ± 0.82 |

Table 9: Accuracy and standard deviation for SVHN over five runs, after processing 200, 400, . . . , 2000 data points (i.e., for the initial training data set and the subsequent 9 batch acquisitions). This table corresponds to Figure 13.

| Method | 200 | 400 | 600 | 800 | 1000 |
|---|---|---|---|---|---|
| SPN-BatchBALD | 18.89 ± 1.25 | 33.90 ± 3.09 | 49.64 ± 4.04 | 56.89 ± 3.41 | 64.91 ± 00.75 |
| BatchBALD | 16.65 ± 2.28 | 33.61 ± 4.7 | 44.15 ± 5.23 | 55.36 ± 1.01 | 61.82 ± 2.37 |
| Random | 15.89 ± 00.37 | 34.18 ± 1.87 | 48.04 ± 1.47 | 55.86 ± 00.20 | 61.72 ± 1.00 |
| Method | 1200 | 1400 | 1600 | 1800 | 2000 |
| SPN-BatchBALD | 70.15 ± 1.12 | 72.82 ± 1.19 | 75.51 ± 1.46 | 76.81 ± 1.13 | 78.46 ± 00.36 |
| BatchBALD | 68.15 ± 2.38 | 71.78 ± 00.51 | 78.85 ± 2.02 | 75.92 ± 1.67 | 77.53 ± 00.89 |
| Random | 67.67 ± 00.38 | 71.83 ± 00.76 | 73.38 ± 1.88 | 75.61 ± 1.74 | 77.86 ± 00.86 |

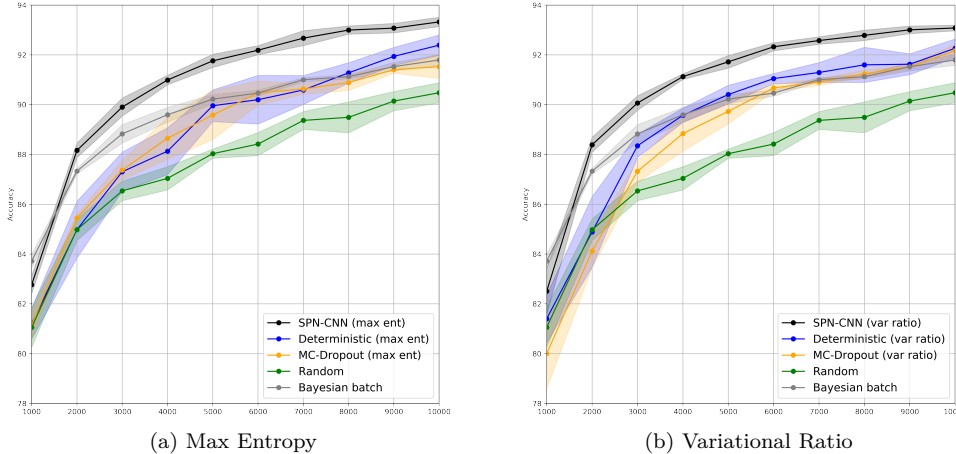

(a) Max Entropy         (b) Variational Ratio

Figure 9: Accuracy results and error bars on the Fashion-MNIST data, averaged over five runs, with std dev indicated.

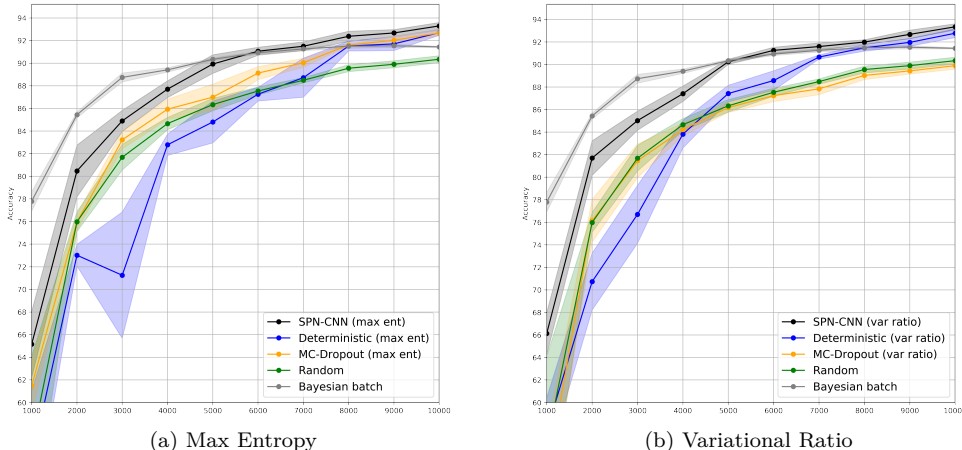

(a) Max Entropy        (b) Variational Ratio

Figure 10: Accuracy results and error bars on the SVHN data, averaged over five runs, with std dev indicated.

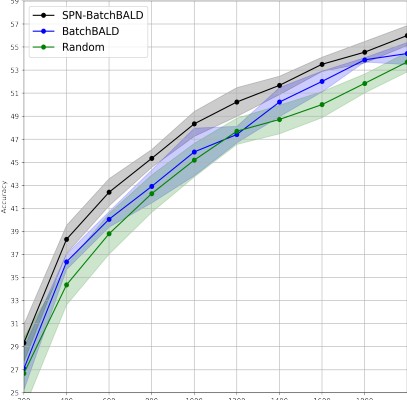

Figure 11: Accuracy on CIFAR-10 with error bars.

- In our code we used part of the code by Gal et al. (2017) for preparing the initial set and new training set in each acquisition.

- We used the code by Kirsch et al. (2019) for computing BatchBALD.

## D  Statistical Tests for CIFAR-10, Fashion-MNIST, and SVHN

Tables 10 through 17 present the $p$ values from our $z$-tests as explained in the main paper.

Regarding the experiments on pure uncertainty-based sampling, we here give two tables for each of the six combinations of dataset (CIFAR-10, Fashion-MNIST, SVHN) and acquisition function (variational ratio, max entropy). In one table, we compare SPN-CNN to MC-Dropout, on the chosen dataset when using the chosen acquisition function; in the other table, we compare SPN-CNN to Bayesian Batch.

Regarding the experiments on mixed-criteria sampling, using SPN-BatchBALD, we provide two tables each for the three datasets (CIFAR-10, Fashion-MNIST, SVHN). In one table, we compare SPN-BatchBALD to BatchBALD on the chosen dataset; in the other table, we compare SPN-BatchBALD to *Random*.

Each table shows $p$-values from paired $z$-tests over all 10,000 test data points. Since we performed five runs of the experiments, each time with a different initial training set, we here report statistical test results for each of these five runs (initial sets 0 through 4), but we mainly focus on the bottom row of each table, labeled

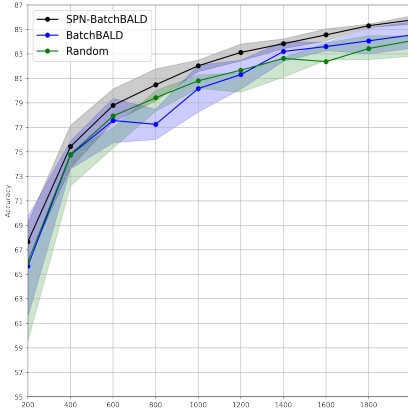

Figure 12: Accuracy on Fashion-MNIST with error bars.

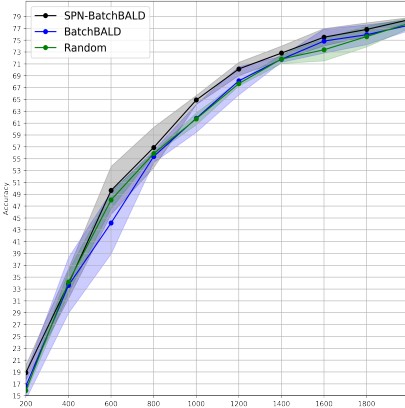

Figure 13: Accuracy on SVHN with error bars.

"mean result". To obtain the $p$ values in this bottom row, we averaged the accuracy values of a method over the five runs. This way, we obtained two mean values (one for SPN-CNN or SPN-BatchBALD, and one for its competitor) that were compared.

| initial dataset | 1000 | 2000 | 3000 | 4000 | 5000 | 6000 | 7000 | 8000 | 9000 | 10000 |
|---|---|---|---|---|---|---|---|---|---|---|
| initial set 0 | 0.00 | 0.00 | 0.05 | 0.00 | 0.00 | 0.00 | 0.00 | 0.00 | 0.00 | 0.00 |
| initial set 1 | 0.60 | 0.00 | 0.00 | 0.00 | 0.00 | 0.00 | 0.00 | 0.00 | 0.00 | 0.00 |
| initial set 2 | 0.00 | 0.00 | 0.00 | 0.00 | 0.00 | 0.00 | 0.00 | 0.00 | 0.00 | 0.00 |
| initial set 3 | 0.00 | 0.01 | 0.01 | 0.00 | 0.00 | 0.01 | 0.00 | 0.00 | 0.00 | 0.00 |
| initial set 4 | 0.00 | 0.00 | 0.00 | 0.00 | 0.14 | 0.00 | 0.00 | 0.00 | 0.00 | 0.00 |
| mean result | 0.00 | 0.00 | 0.00 | 0.00 | 0.05 | 0.01 | 0.00 | 0.01 | 0.00 | 0.00 |

Table 10: $p$ values for five runs on CIFAR-10 (SPN-CNN var ratio vs. MC-Dropout var ratio)

| initial dataset | 1000 | 2000 | 3000 | 4000 | 5000 | 6000 | 7000 | 8000 | 9000 | 10000 |
|---|---|---|---|---|---|---|---|---|---|---|
| initial set 0 | 0.00 | 0.00 | 0.02 | 0.00 | 0.00 | 0.00 | 0.00 | 0.00 | 0.00 | 0.00 |
| initial set 1 | 0.02 | 0.00 | 0.00 | 0.04 | 0.00 | 0.00 | 0.00 | 0.00 | 0.00 | 0.00 |
| initial set 2 | 0.00 | 0.00 | 0.00 | 0.00 | 0.00 | 0.00 | 0.00 | 0.00 | 0.00 | 0.00 |
| initial set 3 | 0.00 | 0.02 | 0.00 | 0.01 | 0.00 | 0.00 | 0.00 | 0.00 | 0.00 | 0.00 |
| initial set 4 | 0.00 | 0.03 | 0.06 | 0.05 | 0.00 | 0.00 | 0.00 | 0.00 | 0.00 | 0.00 |
| mean result | 0.00 | 0.00 | 0.00 | 0.00 | 0.05 | 0.01 | 0.00 | 0.00 | 0.00 | 0.00 |

Table 11: $p$ values for five runs on CIFAR-10 (SPN-CNN max entropy vs. MC-Dropout max entropy)

| initial dataset | 1000 | 2000 | 3000 | 4000 | 5000 | 6000 | 7000 | 8000 | 9000 | 10000 |
|---|---|---|---|---|---|---|---|---|---|---|
| initial set 0 | 0.08 | 0.03 | 0.00 | 0.08 | 0.00 | 0.00 | 0.00 | 0.00 | 0.00 | 0.00 |
| initial set 1 | 0.36 | 0.11 | 0.00 | 0.00 | 0.00 | 0.00 | 0.00 | 0.00 | 0.00 | 0.00 |
| initial set 2 | 0.01 | 0.00 | 0.00 | 0.00 | 0.00 | 0.00 | 0.00 | 0.00 | 0.00 | 0.00 |
| initial set 3 | 0.06 | 0.02 | 0.00 | 0.00 | 0.00 | 0.00 | 0.00 | 0.00 | 0.00 | 0.00 |
| initial set 4 | 0.00 | 0.00 | 0.00 | 0.05 | 0.00 | 0.00 | 0.00 | 0.00 | 0.00 | 0.00 |
| mean result | 0.04 | 0.01 | 0.00 | 0.00 | 0.00 | 0.01 | 0.00 | 0.00 | 0.00 | 0.00 |

Table 12: $p$ values for five runs on CIFAR-10 (SPN-CNN max entropy vs. Bayesian Batch)

| initial dataset | 1000 | 2000 | 3000 | 4000 | 5000 | 6000 | 7000 | 8000 | 9000 | 10000 |
|---|---|---|---|---|---|---|---|---|---|---|
| initial set 0 | 0.06 | 0.02 | 0.12 | 0.00 | 0.00 | 0.00 | 0.00 | 0.00 | 0.00 | 0.00 |
| initial set 1 | 0.07 | 0.00 | 0.00 | 0.00 | 0.00 | 0.00 | 0.00 | 0.00 | 0.00 | 0.00 |
| initial set 2 | 0.00 | 0.00 | 0.00 | 0.00 | 0.00 | 0.00 | 0.00 | 0.00 | 0.00 | 0.00 |
| initial set 3 | 0.00 | 0.00 | 0.10 | 0.00 | 0.00 | 0.00 | 0.00 | 0.00 | 0.00 | 0.00 |
| initial set 4 | 0.00 | 0.00 | 0.00 | 0.00 | 0.00 | 0.00 | 0.00 | 0.00 | 0.00 | 0.00 |
| mean result | 0.05 | 0.01 | 0.00 | 0.00 | 0.00 | 0.01 | 0.00 | 0.00 | 0.00 | 0.00 |

Table 13: $p$ values for five runs on CIFAR-10 (SPN-CNN var ratio vs. Bayesian Batch)

| initial dataset | 1000 | 2000 | 3000 | 4000 | 5000 | 6000 | 7000 | 8000 | 9000 | 10000 |
|---|---|---|---|---|---|---|---|---|---|---|
| initial set 0 | 0.36 | 0.00 | 0.00 | 0.01 | 0.00 | 0.00 | 0.00 | 0.00 | 0.00 | 0.09 |
| initial set 1 | 0.00 | 0.00 | 0.00 | 0.00 | 0.00 | 0.00 | 0.00 | 0.00 | 0.00 | 0.01 |
| initial set 2 | 0.00 | 0.00 | 0.00 | 0.00 | 0.00 | 0.00 | 0.00 | 0.00 | 0.00 | 0.00 |
| initial set 3 | 0.02 | 0.00 | 0.00 | 0.00 | 0.00 | 0.00 | 0.00 | 0.00 | 0.00 | 0.02 |
| initial set 4 | 0.00 | 0.00 | 0.00 | 0.00 | 0.00 | 0.00 | 0.00 | 0.00 | 0.00 | 0.00 |
| mean result | 0.02 | 0.00 | 0.00 | 0.00 | 0.00 | 0.00 | 0.00 | 0.00 | 0.00 | 0.01 |

Table 14: $p$ values for five runs on Fashion-MNIST (SPN-CNN var ratio vs. MC-Dropout var ratio)

| initial dataset | 1000 | 2000 | 3000 | 4000 | 5000 | 6000 | 7000 | 8000 | 9000 | 10000 |
|---|---|---|---|---|---|---|---|---|---|---|
| initial set 0 | 0.36 | 0.00 | 0.00 | 0.00 | 0.00 | 0.01 | 0.00 | 0.00 | 0.00 | 0.00 |
| initial set 1 | 0.04 | 0.00 | 0.00 | 0.00 | 0.01 | 0.00 | 0.00 | 0.00 | 0.00 | 0.01 |
| initial set 2 | 0.00 | 0.00 | 0.00 | 0.00 | 0.00 | 0.00 | 0.00 | 0.00 | 0.00 | 0.00 |
| initial set 3 | 0.00 | 0.00 | 0.00 | 0.00 | 0.00 | 0.00 | 0.00 | 0.00 | 0.00 | 0.00 |
| initial set 4 | 0.00 | 0.00 | 0.00 | 0.00 | 0.00 | 0.00 | 0.00 | 0.00 | 0.00 | 0.00 |
| mean result | 0.00 | 0.00 | 0.00 | 0.00 | 0.00 | 0.00 | 0.00 | 0.00 | 0.00 | 0.00 |

Table 15: $p$ values for five runs on Fashion-MNIST (SPN-CNN max entropy vs. MC-Dropout max entropy)

| initial dataset | 1000 | 2000 | 3000 | 4000 | 5000 | 6000 | 7000 | 8000 | 9000 | 10000 |
|---|---|---|---|---|---|---|---|---|---|---|
| initial set 0 | 0.92 | 0.00 | 0.00 | 0.00 | 0.00 | 0.01 | 0.00 | 0.00 | 0.00 | 0.00 |
| initial set 1 | 0.99 | 0.24 | 0.01 | 0.00 | 0.01 | 0.00 | 0.00 | 0.00 | 0.00 | 0.00 |
| initial set 2 | 0.45 | 0.03 | 0.00 | 0.00 | 0.00 | 0.00 | 0.00 | 0.00 | 0.00 | 0.00 |
| initial set 3 | 0.95 | 0.05 | 0.01 | 0.00 | 0.00 | 0.00 | 0.00 | 0.00 | 0.00 | 0.00 |
| initial set 4 | 0.99 | 0.01 | 0.37 | 0.02 | 0.00 | 0.00 | 0.00 | 0.00 | 0.00 | 0.00 |
| mean result | 0.96 | 0.04 | 0.01 | 0.00 | 0.00 | 0.00 | 0.00 | 0.00 | 0.00 | 0.00 |

Table 16: $p$ values for five runs on Fashion-MNIST (SPN-CNN max entropy vs. Bayesian Batch)

| initial dataset | 1000 | 2000 | 3000 | 4000 | 5000 | 6000 | 7000 | 8000 | 9000 | 10000 |
|---|---|---|---|---|---|---|---|---|---|---|
| initial set 0 | 0.99 | 0.00 | 0.00 | 0.00 | 0.00 | 0.01 | 0.00 | 0.00 | 0.00 | 0.00 |
| initial set 1 | 0.76 | 0.03 | 0.01 | 0.00 | 0.00 | 0.00 | 0.00 | 0.00 | 0.00 | 0.00 |
| initial set 2 | 0.66 | 0.01 | 0.00 | 0.00 | 0.01 | 0.00 | 0.00 | 0.00 | 0.00 | 0.00 |
| initial set 3 | 0.99 | 0.07 | 0.01 | 0.00 | 0.00 | 0.00 | 0.00 | 0.00 | 0.00 | 0.00 |
| initial set 4 | 0.99 | 0.00 | 0.01 | 0.00 | 0.00 | 0.00 | 0.00 | 0.00 | 0.00 | 0.00 |
| mean result | 0.98 | 0.01 | 0.00 | 0.00 | 0.00 | 0.00 | 0.00 | 0.00 | 0.00 | 0.00 |

Table 17: $p$ values for five runs on Fashion-MNIST (SPN-CNN var ratio vs. Bayesian Batch)

| initial dataset | 1000 | 2000 | 3000 | 4000 | 5000 | 6000 | 7000 | 8000 | 9000 | 10000 |
|---|---|---|---|---|---|---|---|---|---|---|
| initial set 0 | 0.00 | 0.00 | 0.00 | 0.97 | 0.00 | 0.00 | 0.00 | 0.00 | 0.00 | 0.00 |
| initial set 1 | 0.00 | 0.00 | 0.08 | 0.00 | 0.00 | 0.00 | 0.00 | 0.00 | 0.00 | 0.00 |
| initial set 2 | 0.00 | 0.00 | 0.00 | 0.00 | 0.00 | 0.00 | 0.00 | 0.00 | 0.00 | 0.00 |
| initial set 3 | 0.00 | 0.90 | 0.00 | 0.00 | 0.00 | 0.00 | 0.00 | 0.00 | 0.00 | 0.00 |
| initial set 4 | 0.00 | 0.01 | 0.00 | 0.00 | 0.00 | 0.00 | 0.00 | 0.00 | 0.00 | 0.00 |
| mean result | 0.00 | 0.00 | 0.00 | 0.00 | 0.00 | 0.00 | 0.00 | 0.00 | 0.00 | 0.00 |

Table 18: $p$ values for five runs on SVHN (SPN-CNN var ratio vs. MC Dropout var ratio)

| initial dataset | 1000 | 2000 | 3000 | 4000 | 5000 | 6000 | 7000 | 8000 | 9000 | 10000 |
|---|---|---|---|---|---|---|---|---|---|---|
| initial set 0 | 0.00 | 0.00 | 0.00 | 0.97 | 0.00 | 0.00 | 0.00 | 0.00 | 0.00 | 0.00 |
| initial set 1 | 0.00 | 0.00 | 0.08 | 0.00 | 0.00 | 0.00 | 0.00 | 0.00 | 0.00 | 0.00 |
| initial set 2 | 0.00 | 0.00 | 0.00 | 0.00 | 0.00 | 0.00 | 0.00 | 0.00 | 0.00 | 0.00 |
| initial set 3 | 0.00 | 0.90 | 0.00 | 0.00 | 0.00 | 0.00 | 0.00 | 0.00 | 0.00 | 0.00 |
| initial set 4 | 0.00 | 0.01 | 0.00 | 0.00 | 0.00 | 0.00 | 0.00 | 0.00 | 0.00 | 0.00 |
| mean result | 0.00 | 0.00 | 0.00 | 0.00 | 0.00 | 0.00 | 0.00 | 0.00 | 0.00 | 0.00 |

Table 19: $p$ values for five runs on SVHN (SPN-CNN max entropy vs. MC Dropout max entropy)

| initial dataset | 1000 | 2000 | 3000 | 4000 | 5000 | 6000 | 7000 | 8000 | 9000 | 10000 |
|---|---|---|---|---|---|---|---|---|---|---|
| initial set 0 | 1.00 | 0.90 | 0.90 | 0.84 | 0.64 | 0.22 | 0.13 | 0.01 | 0.00 | 0.00 |
| initial set 1 | 1.00 | 0.90 | 1.00 | 0.90 | 0.51 | 0.03 | 0.01 | 0.13 | 0.00 | 0.00 |
| initial set 2 | 1.00 | 0.90 | 1.00 | 0.90 | 0.82 | 0.18 | 0.25 | 0.04 | 0.00 | 0.00 |
| initial set 3 | 1.00 | 1.00 | 0.9 | 0.45 | 0.27 | 0.50 | 0.04 | 0.00 | 0.00 | 0.00 |
| initial set 4 | 1.00 | 1.00 | 0.90 | 0.90 | 0.44 | 0.50 | 0.36 | 0.07 | 0.10 | 0.00 |
| mean result | 1.00 | 0.90 | 0.90 | 0.90 | 0.58 | 0.20 | 0.12 | 0.00 | 0.00 | 0.00 |

Table 20: $p$ values for five runs on SVHN (SPN-CNN var ratio vs. Bayesian Batch)

| initial dataset | 1000 | 2000 | 3000 | 4000 | 5000 | 6000 | 7000 | 8000 | 9000 | 10000 |
|---|---|---|---|---|---|---|---|---|---|---|
| initial set 0 | 1.00 | 1.00 | 1.00 | 1.00 | 0.90 | 0.91 | 0.52 | 0.00 | 0.00 | 0.00 |
| initial set 1 | 1.00 | 1.00 | 1.00 | 0.99 | 0.99 | 0.81 | 0.05 | 0.00 | 0.00 | 0.00 |
| initial set 2 | 1.00 | 1.00 | 1.00 | 0.99 | 0.99 | 0.35 | 0.40 | 0.00 | 0.00 | 0.00 |
| initial set 3 | 1.00 | 1.00 | 0.99 | 0.99 | 0.99 | 0.80 | 0.36 | 0.00 | 0.00 | 0.00 |
| initial set 4 | 1.00 | 1.00 | 0.99 | 0.99 | 0.98 | 0.72 | 0.80 | 0.00 | 0.00 | 0.00 |
| mean result | 1.00 | 1.00 | 1.00 | 0.99 | 0.99 | 0.85 | 0.42 | 0.01 | 0.00 | 0.00 |

Table 21: $p$ values for five runs on SVHN (SPN-CNN max entropy vs. Bayesian Batch)

| initial dataset | 200 | 400 | 600 | 800 | 1000 | 1200 | 1400 | 1600 | 1800 | 2000 |
|---|---|---|---|---|---|---|---|---|---|---|
| initial set 0 | 0.00 | 0.03 | 0.00 | 0.00 | 0.00 | 0.01 | 0.00 | 0.00 | 0.16 | 0.00 |
| initial set 1 | 0.75 | 0.00 | 0.00 | 0.02 | 0.27 | 0.00 | 0.10 | 0.05 | 0.11 | 0.03 |
| initial set 2 | 0.00 | 0.02 | 0.00 | 0.43 | 0.00 | 0.00 | 0.35 | 0.00 | 0.00 | 0.44 |
| initial set 3 | 0.02 | 0.00 | 0.05 | 0.00 | 0.01 | 0.00 | 0.30 | 0.52 | 0.12 | 0.02 |
| initial set 4 | 0.00 | 0.00 | 0.00 | 0.00 | 0.00 | 0.00 | 0.00 | 0.00 | 0.16 | 0.00 |
| mean result | 0.00 | 0.00 | 0.00 | 0.00 | 0.00 | 0.00 | 0.02 | 0.02 | 0.17 | 0.01 |

Table 22: $p$ values for five runs on CIFAR-10 (SPN-BatchBALD vs. BatchBALD)

| initial dataset | 200 | 400 | 600 | 800 | 1000 | 1200 | 1400 | 1600 | 1800 | 2000 |
|---|---|---|---|---|---|---|---|---|---|---|
| initial set 0 | 0.00 | 0.00 | 0.00 | 0.00 | 0.00 | 0.00 | 0.01 | 0.00 | 0.00 | 0.00 |
| initial set 1 | 0.96 | 0.00 | 0.02 | 0.01 | 0.06 | 0.00 | 0.01 | 0.01 | 0.00 | 0.00 |
| initial set 2 | 0.20 | 0.00 | 0.00 | 0.02 | 0.00 | 0.00 | 0.00 | 0.00 | 0.00 | 0.00 |
| initial set 3 | 0.00 | 0.00 | 0.00 | 0.01 | 0.00 | 0.00 | 0.01 | 0.00 | 0.00 | 0.00 |
| initial set 4 | 0.00 | 0.00 | 0.00 | 0.00 | 0.00 | 0.06 | 0.00 | 0.00 | 0.00 | 0.03 |
| mean result | 0.00 | 0.00 | 0.00 | 0.00 | 0.00 | 0.00 | 0.02 | 0.02 | 0.17 | 0.01 |

Table 23: $p$ values for five runs on CIFAR-10 (SPN-BatchBALD vs. *Random*)

| initial dataset | 200 | 400 | 600 | 800 | 1000 | 1200 | 1400 | 1600 | 1800 | 2000 |
|---|---|---|---|---|---|---|---|---|---|---|
| initial set 0 | 0.00 | 0.00 | 0.44 | 0.00 | 0.09 | 0.00 | 0.50 | 0.01 | 0.00 | 0.03 |
| initial set 1 | 0.00 | 0.00 | 0.03 | 0.00 | 0.00 | 0.00 | 0.00 | 0.24 | 0.01 | 0.00 |
| initial set 2 | 0.00 | 0.20 | 0.15 | 0.00 | 0.00 | 0.70 | 0.23 | 0.16 | 0.02 | 0.02 |
| initial set 3 | 0.00 | 0.99 | 0.00 | 0.00 | 0.00 | 0.00 | 0.08 | 0.00 | 0.00 | 0.00 |
| initial set 4 | 0.00 | 0.90 | 0.00 | 0.00 | 0.00 | 0.06 | 0.43 | 0.16 | 0.07 | 0.01 |
| mean result | 0.00 | 0.16 | 0.02 | 0.00 | 0.00 | 0.00 | 0.15 | 0.03 | 0.00 | 0.01 |

Table 24: $p$ values for five runs on Fashion-MNIST (SPN-BatchBALD vs. BatchBALD)

| initial dataset | 200 | 400 | 600 | 800 | 1000 | 1200 | 1400 | 1600 | 1800 | 2000 |
|---|---|---|---|---|---|---|---|---|---|---|
| initial set 0 | 0.00 | 0.00 | 0.92 | 0.97 | 0.07 | 0.62 | 0.00 | 0.00 | 0.00 | 0.00 |
| initial set 1 | 0.06 | 0.04 | 0.04 | 0.03 | 0.09 | 0.00 | 0.00 | 0.00 | 0.00 | 0.01 |
| initial set 2 | 0.00 | 0.38 | 0.15 | 0.01 | 0.00 | 0.00 | 0.25 | 0.02 | 0.00 | 0.35 |
| initial set 3 | 0.00 | 0.90 | 0.00 | 0.11 | 0.00 | 0.00 | 0.34 | 0.17 | 0.00 | 0.00 |
| initial set 4 | 0.00 | 0.51 | 0.00 | 0.00 | 0.00 | 0.00 | 0.00 | 0.00 | 0.00 | 0.00 |
| mean result | 0.00 | 0.18 | 0.06 | 0.03 | 0.00 | 0.02 | 0.00 | 0.00 | 0.00 | 0.00 |

Table 25: $p$ values for five runs on Fashion-MNIST (SPN-BatchBALD vs. *Random*)

| initial dataset | 200 | 400 | 600 | 800 | 1000 | 1200 | 1400 | 1600 | 1800 | 2000 |
|---|---|---|---|---|---|---|---|---|---|---|
| initial set 0 | 0.00 | 0.00 | 0.00 | 0.00 | 0.00 | 0.01 | 0.00 | 0.12 | 0.04 | 0.00 |
| initial set 1 | 0.99 | 1.00 | 1.00 | 0.00 | 0.00 | 0.00 | 0.00 | 0.00 | 0.00 | 0.51 |
| initial set 2 | 0.00 | 1.00 | 1.00 | 0.99 | 0.99 | 0.35 | 0.40 | 0.00 | 0.00 | 0.00 |
| initial set 3 | 0.00 | 0.61 | 0.00 | 0.00 | 0.00 | 0.00 | 0.00 | 0.00 | 0.00 | 0.07 |
| initial set 4 | 0.00 | 0.80 | 0.00 | 0.00 | 0.00 | 0.08 | 0.09 | 0.22 | 0.00 | 0.09 |
| mean result | 0.00 | 0.50 | 0.00 | 0.00 | 0.00 | 0.00 | 0.42 | 0.04 | 0.00 | 0.05 |

Table 26: $p$ values for five runs on SVHN (SPN-BatchBALD vs. BatchBALD)

| initial dataset | 200 | 400 | 600 | 800 | 1000 | 1200 | 1400 | 1600 | 1800 | 2000 |
|---|---|---|---|---|---|---|---|---|---|---|
| initial set 0 | 0.00 | 0.93 | 0.00 | 0.00 | 0.00 | 0.01 | 0.50 | 0.12 | 0.04 | 0.00 |
| initial set 1 | 0.00 | 1.00 | 0.00 | 0.00 | 0.00 | 0.00 | 0.00 | 0.20 | 0.13 | 0.00 |
| initial set 2 | 0.00 | 0.99 | 0.00 | 0.00 | 0.0 | 0.00 | 0.00 | 0.00 | 0.00 | 0.77 |
| initial set 3 | 0.00 | 0.55 | 0.00 | 0.00 | 0.00 | 0.00 | 0.50 | 0.00 | 0.00 | 0.00 |
| initial set 4 | 0.00 | 0.99 | 0.00 | 0.00 | 0.00 | 0.00 | 0.28 | 0.22 | 0.00 | 0.00 |
| mean result | 0.00 | 0.46 | 0.00 | 0.00 | 0.00 | 0.00 | 0.42 | 0.04 | 0.00 | 0.07 |

Table 27: $p$ values for five runs on SVHN (SPN-BatchBALD vs. *Random*)

