# OpenReview forum: "Using Sum-Product Networks to Assess Uncertainty in Deep Active Learning"
_TMLR — Accepted by TMLR_

### Review · Reviewer_4wzp · 2023-10-30

**Summary Of Contributions:**

The paper aims to present a new approach for computing uncertainty in deep active learning applied to Convolutional Neural Networks. The authors propose leveraging the feature representation obtained from the CNN as data for training a Sum-Product Network (SPN). Given that SPNs are commonly employed for estimating data distributions, the authors suggest that they are apt for estimating class probabilities in the context of deep active learning.

**Audience:**

No

**Broader Impact Concerns:**

At present, there's no clear indication of the broader ethical implications or potential misuse of the proposed methodology. It would be beneficial for the authors to include a Broader Impact Statement addressing these aspects.

**Claims And Evidence:**

Yes

**Requested Changes:**

- Critical: The authors must address the method's inability to handle datasets with a large number of categories. If the proposed approach inherently has this limitation, the paper should discuss potential ways to overcome or mitigate it.

- Critical: A comprehensive revision of the writing is essential. Grammatical mistakes and unclear sentences must be rectified to ensure the paper communicates its ideas effectively.

- Recommended: While it's understood that the paper aims to employ SPN in the context of deep active learning, further innovation or enhancement to the base SPN methodology would greatly increase the paper's significance.

**Strengths And Weaknesses:**

### Strengths:

- The paper introduces a unique perspective by integrating SPNs into the deep active learning process, showcasing an intersection of two significant machine learning methodologies.

### Weaknesses:

- The methodology primarily hinges on a straightforward application of the SPN without any further enhancement, leading to a lack of innovation.
- The proposed method seems to be not applicable or scalable to datasets with a large number of categories, thereby limiting its practical applicability.
- The writing and presentation of the paper are not clear. There are numerous grammatical errors and convoluted explanations, making it challenging for readers to grasp the content.

---

### Review · Reviewer_BFDo · 2023-11-08

**Summary Of Contributions:**

The paper introduces a framework for using a Sum-Product Network (SPN) for computing the uncertainty in deep active learning systems that use Convolutional Neural Networks (CNNs). The core contribution is leveraging the feature representations generated by the CNN to train a Sum-Product Network (SPN), which is known to be good at estimating data distributions. By doing so, the paper claims that SPNs can effectively estimate class probabilities, which are crucial for standard acquisition function used in active learning. The proposed method is validated through experiments on multiple standard image classification benchmark datasets, where it is compared with several uncertainty-based active learning methods.

**Audience:**

No

**Broader Impact Concerns:**

None.

**Claims And Evidence:**

Yes

**Requested Changes:**

See the weakness above.

**Strengths And Weaknesses:**

**Strength**

1. Simple yet effective method
2. Extensive experiments on multiple benchmark datasets

**Weakness**

1. The Introduction section could benefit from a more concise structure. While the flow up to paragraph 6 is clear, the subsequent sections become harder to convey. Are paragraph 7 and onwards the related works of the paper?

2. The primary contributions appear to be an application of existing uncertainty estimation techniques using Sum Product Networks (SPN) to the domain of uncertainty-based active learning. This approach may be perceived as an incremental contribution. Additionally, the concept of incorporating a separate layer before the softmax for enhanced prediction uncertainty is well-established in literature (e.g., Temperature scaling [1], Platt Scaling [1], Dirichlet calibration [2], DUC [3], etc.).

*reference*

[1] https://proceedings.mlr.press/v70/guo17a/guo17a.pdf

[2] https://proceedings.neurips.cc/paper/2019/file/8ca01ea920679a0fe3728441494041b9-Paper.pdf

[3] https://openreview.net/forum?id=4WM4cy42B81

3. Since SPN is being used to better capture the uncertainty of a prediction, it will be good to see the Expected Calibration Error (ECE) of the predictions.

4. An essential criterion for evaluating uncertainty-based research is its effectiveness on out-of-distribution (OOD) samples, which is crucial for practical applications. For example, many calibration methods that perform well on in-distribution data fail with OOD samples. It would be advantageous for the paper to address the proposed method's performance in OOD scenarios, such as active domain adaptation, to demonstrate its potential for real-world deployment.

---

> ### Author Response · Authors · 2023-12-17
> **Response to Review**
>
> Thank you for your detailed and helpful review of our paper. We will respond to your concerns item by item.
>
> 1. Yes, paragraph 7 onwards discussed related work. We agree that the introduction will benefit from having a more clear structure. This is very easy to do in a revision.
> 2. The main purpose of our research was to show that training SPN on the feature representation of CNN yields methods that improve the state-of-the-art in uncertainty estimation in deep learning -- we believe that we successfully demonstrated this. Our method is simple; whether or not this means the significant of our contribution is low is in our eyes rather arguable. Either way, it is our understanding that TMLR's mission is to value technical correctness over significant, the latter being a subjective matter.
> 3. We have provided two types of experiments, following the experimental design in other recent papers in leading venues, such as NeurIPS, ICML, Machine Learning. In the first type of experiment, we test our approach to estimating uncertainty INDIRECTLY, i.e., we try to answer the question how accurately deep active learning with uncertainty sampling makes predictions when existing uncertainty methods are replaced by ours. To answer this question, ECE is actually not relevant. In the second type of experiment, we try to answer the question whether our method ends up producing rather random values when applied on data from a completely unseen distribution (as it should). Again, we do not really see a need for ECE evaluation here. If the reviewers/associate editor consider ECE tests as the critical part that decides between accepting and rejecting the paper, we will be happy to run these experiments, yet we don't foresee them as being able to change the message of our paper in any way, irrespective of the outcome.
> 4. In fact, our experiments reported in Figure 5 are a kind of OOD evaluation of our uncertainty estimates. If the reviewer means OOD evaluation of a deep active learner that employs our uncertainty estimation method, we would like to stress that this is somewhat out of scope for our paper. The purpose of our paper is NOT to fine-tune/design the absolute best method for deep active learning, but to show that SPNs trained on features extracted by CNNs yield a very promising alternative to state-of-the-art uncertainty estimators. We believe that our work provides crucial insights into uncertainty estimation, which may inform future work along these lines.

---

### Review · Reviewer_z1Nw · 2023-12-04

**Summary Of Contributions:**

This work aims to improve deep active learning through designing a better acquisition function, which is used to evaluate the uncertainty of unlabeled data. To do so, the authors trained SPNs  (Sum-Product-Network) on trained CNNs and used the outputs of SPNs to evaluate the uncertainty of data. This new method was compared to an earlier method that evaluated the uncertainty through MC-Dropout and the authors demonstrated the effectiveness of their method on multiple standard benchmarks.

**Audience:**

Yes

**Broader Impact Concerns:**

No clear negative society impacts from this.

**Claims And Evidence:**

Yes

**Requested Changes:**

See the weakness.

**Strengths And Weaknesses:**

**Strengths:**

The empirical evaluation results presented in this paper indicate that the new method of estimating uncertainty works well. The authors have also evaluated their method on multiple benchmarks to show the robustness of the strength of their method. Different ways of active learning are also tested and the proposed method seems to outperform existing methods in both pure uncertainty-based sampling and mixed sampling methods.

**Weaknesses:**

1.	Although the authors presented many empirical evaluation results, it is unclear to me whether some of these evaluation results, especially those on existing methods, are consistent with what have been published in literature. Can the authors compare to earlier results in an apple-to-apple comparison to show that their benefit is not because of certain implementation issues in existing methods?

2.	Ablation studies are missing to help better understand this method. The use of SPN to predict uncertainty indeed seems to be beneficial, but how about using different dense layers of CNN to train SPN? What about other potential architectures like SPN to predict uncertainty? More analyses can also be useful to help understand the method. For example, is the uncertainty estimated by SPN totally different from MC-Dropout? If not, what is their correlation?


3.	The writing and figures need to be improved. For example, the first section is not well structured and too long. All the figures are also longer than the text limit and the axis tick values are really too small to be seen.

---

### Decision · Action_Editor_bNMn · 2024-01-24

**Recommendation:** Accept with minor revision

**Comment:**

The decision to accept is based on a technically sound method to estimating uncertainty and its usage in active learning as well is its extensive evaluation. Reviewers raised various concerns and authors should address them. Specifically,
- Add an analysis on calibration (ECE)
- Improve the writing
- Add a discussion on SPN and discuss engineering decisions (eg. which layer to use etc.)
- Add a discussion on limitations, such as scalability to large category datasets

**Audience:**

The findings of this paper are interesting to the community of active learning and uncertainty estimation. Although the novelty is limited, the relevant community would find the paper interesting.

**Claims And Evidence:**

The manuscript is about an deep active learning by using Sum-Product Networks (SPNs) with Convolutional Neural Networks (CNNs) to estimate the uncertainty of unlabeled data. The authors have provided convincing evidence through extensive empirical evaluations on multiple standard benchmarks to support their claims. These evaluations demonstrate the effectiveness of their method compared to existing methods. Despite various concerns raised by the reviewers, the evidence as it stands is accurate and clear, supporting the main claims of the paper. The issues raised by the reviewers are mostly about novelty and additional studies which I believe do not impact the evidence of the claims.